# Deciphering direct transcriptional effects of epigenetic compounds through large-scale new RNA profiling

Leonard Hartmanis [1], Daniel Ramsköld [1], Gert-Jan Hendriks [1], Per Johnsson[1], Gustav Hallén [1,2], Ran Ma[3], Anton J. M. Larsson [1], Salomé Hahne[1], Christoph Ziegenhain [1], Johan Hartman [3,4] & Rickard Sandberg [1] ✉

Examining direct transcriptional effects of genetic and chemical perturbations is crucial for understanding gene expression mechanisms. Standard RNA-seq experiments often overlook these direct effects, and current methods for profiling nascent RNA are usually time-consuming. Here, we adapted single-cell 4sU-based sequencing into a scalable, automated mini-bulk format to profile new RNA in smaller cell populations. This approach enabled us to map the direct transcriptional effects of epigenetic regulators. Brief exposure to SAHA (an HDAC inhibitor) revealed hundreds of directly responsive genes, many showing altered transcriptional bursting kinetics, with promoter regions enriched in binding sites for factors including bromodomain proteins. Profiling 83 epigenetic compounds uncovered direct transcriptional impacts from inhibitors of bromodomain proteins, histone deacetylases, and histone demethylases. Notably, chemically similar HDAC inhibitors elicited concordant direct responses and intronic expression analyses mirrored the direct effects seen in new RNA. This work highlights powerful approaches for investigating transcriptional mechanisms.

Genetic and chemical perturbations are widely employed to explore gene expression regulation in health and disease. Small molecules that target enzymes involved in reading, writing, or erasing epigenetic modifications are increasingly finding their way into clinical applications[1]. Modulating chromatin-modifying enzymes affects a broad array of target genes, but our understanding of the molecular mechanisms and specificities of these compounds remains incomplete. A significant challenge in studying the effects of these compounds is the limited ability of conventional RNA sequencing experiments to map direct effects, as illustrated in Fig. 1a. Standard RNA-seq analyses lack the resolution to detect immediate, compound-specific transcriptional responses following short treatment durations, because the more abundant pre-existing mRNA molecules transcribed before compound administration obscure the compound-induced signals. Consequently, most standard RNA-seq analyses have concentrated on the transcriptional effects observable after extended treatment periods. For instance, systematic transcriptome-based compound screens on cells have typically examined effects noticeable at 6 or 24 h post-compound administration[2-4]. These studies have provided valuable insights into the long-term, downstream phenotypic consequences of compound treatments (Fig. 1a), but they lack the resolution necessary to study direct effects on transcription.

Pioneering studies have shown that the direct transcriptional effects of pharmacological or genetic perturbations can be observed in nascent RNA within minutes to hours[5-7]. Although several methods are available for profiling nascent or newly transcribed RNA[7,8], they are often time-consuming, limited in scale, or require large amounts of input material. To address these challenges, we and others have

[1]Department of Cell and Molecular Biology, Karolinska Institutet, Stockholm, Sweden. [2]Department of Clinical Chemistry, Karolinska University Hospital, Stockholm, Sweden. [3]Department of Oncology-Pathology, Karolinska Institutet, Stockholm, Sweden. [4]Department of Clinical Pathology and Cancer Diagnostics, Karolinska University Hospital, Stockholm, Sweden. ✉e-mail: Rickard.Sandberg@ki.se

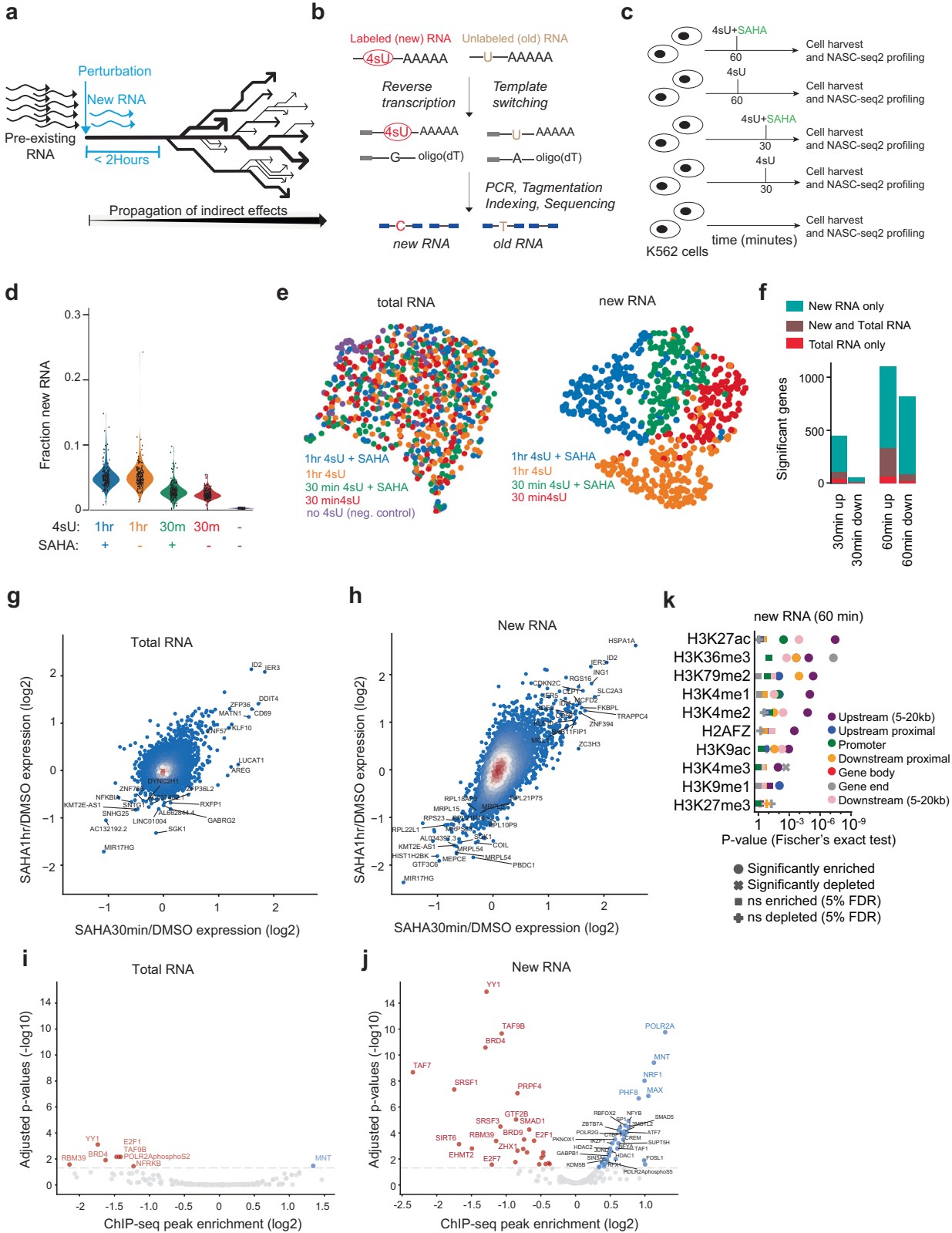

developed methods that combine single-cell RNA sequencing with nucleotide analogs to label newly transcribed mRNA molecules (Fig. 1b). These methods enable the detection of direct response genes[9–12]. Our specific approach builds on SLAM seq chemistry[6], which allows for the in silico separation of labeled (new) and unlabeled (old) RNA by inducing sequence errors during the reverse transcription of labeled RNA molecules.

In this study, we explored the direct transcriptional effects of compounds targeting epigenetic enzymes. Our findings demonstrate that single-cell profiling of new RNA can reveal these effects as early as 30 min after compound administration. Additionally, we developed a high-throughput strategy for new RNA profiling, allowing us to examine cells exposed to over 80 different compounds that affect epigenetic processes. This approach enables the

**Fig. 1 | Revealing direct effects of SAHA by single-cell new RNA-sequencing.**
**a** Illustration of how direct transcriptional effects of perturbations manifest rapidly, followed by propagation of indirect effects. **b** Illustrating 4sU incorporation into newly transcribed RNA, whereas pre-existing RNA remains unlabeled. Alkylation of 4sU bases results in the misincorporation of guanines (instead of adenines) during reverse transcription. The 4sU-induced mismatches are detected in the RNA-seq reads to distinguish newly transcribed (new RNA) from pre-existing RNA. **c** Experimental overview of NASC-seq2 experiment in K562. Cells were treated with SAHA and 4sU for 30 min or 1 h, and compared to control cells exposed to 4sU only or neither compound. **d** Violin plots summarizing the fractions of new RNA per cell, separately for each condition. **e** UMAP visualization of single-cell expression profiles, colored by experimental condition. Left: analysis of total (new + pre-existing) RNA per cell; right: analysis of new RNA per cell. **f** Bar plots showing the number of significant genes detected after 30 min and 1 h of treatment, stratified by up- and down-regulation, and by total and new RNA analysis.

**g** Scatter plot showing the effect of SAHA treatment for 30 min (x-axis) and 1 h (y-axis), based on total RNA. **h** Scatter plot showing the effect of SAHA treatment for 30 min (x-axis) and 1 h (y-axis), based on new RNA. **i** Volcano plot showing factors with enriched ChIP-seq binding in promoters of significantly differentially expressed genes based on total RNA after 1 h SAHA treatment. **j** Volcano plot showing factors with enriched ChIP-seq binding in promoters of significantly differentially expressed genes based on new RNA after 1 h SAHA treatment. **i, j** Enrichments were evaluated using Fisher's exact test (two-sided), with the resulting P values adjusted using the Benjamini-Hochberg procedure. **k** Enrichment of histone marks near SAHA-responsive genes (1 h new RNA), based on ChIP-seq data from untreated K562 cells. Color indicates genomic region relative to gene, shape indicates significance (with or without multiple testing correction), and the x-axis the nominal *P*-values (Fishers exact test, two-sided). Source data are provided as a Source data file.

identification of transcriptional patterns associated with various compound classes.

## Results

We initially examined the direct transcriptional effects of the histone deacetylase (HDAC) inhibitor suberoylanilide hydroxamic acid (SAHA), the first HDAC inhibitor approved by the FDA for the treatment of neoplastic disease[13]. To this end, we treated K562 cells with SAHA (10 μM) and 4sU for either 30 or 60 min and utilized NASC-seq2[14] to label the polyA+ RNA transcribed during the 4sU labeling period (hereafter referred to as new RNA) (Fig. 1c). Additionally, we prepared control libraries of K562 cells exposed to 4sU alone for 30 or 60 min, as well as libraries from cells cultured without 4sU or SAHA (Fig. 1c). After performing quality control, approximately 340 cells per condition were included for downstream analysis (e.g., 341 and 344 cells, for 30- and 60-min SAHA+4sU treatment, respectively) (Supplementary Fig. 1a–c). As expected, we observed an increase in new RNA with longer labeling times, averaging 3% new RNA after 30 min 4sU labeling and 6% after 60 min (Fig. 1d). The addition of SAHA did not affect the total amount of new RNA detected at these time points (Fig. 1d), indicating that SAHA treatment for 30 and 60 min did not have an immediate global effect on transcriptional activity in K562 cells. Interestingly, UMAP projection of cells to two dimensions using total RNA profiles (i.e., without distinguishing between pre-existing and new RNA) and new RNA profiles yielded markedly different results. Analysis of total RNA profiles showed no separation of cells by treatment conditions. Conversely, the analysis of new RNA profiles revealed a clear separation based on SAHA and 4sU exposure (Fig. 1e). This discrepancy was not due to lower variation among new RNA profiles (Supplementary Fig. 1d). Rather, the new RNA profiles consistently detected SAHA effects on transcription that remained hidden when analyzing the whole transcriptomes of the cells. To identify the genes significantly up- and down-regulated by SAHA treatment, we used DESeq2[15] on SAHA-treated cells compared to control cells exposed only to 4sU, analyzing total and new RNA separately. The analysis of new RNA revealed a large number of significantly differentially expressed genes, with 455 and 1,834 genes identified for the 30- and 60-min SAHA treatments, respectively (False Discovery Rate < 0.05). This is a significant increase compared to the analysis of total RNA, which identified 121 and 414 genes under the same conditions (Fig. 1f, Supplementary Fig. 1e and Supplemental Data 1). Interestingly, the initial response to SAHA after 30 min predominantly showed an upregulation of genes, while the 60-min response exhibited more balanced levels of upregulation and downregulation (Fig. 1f). Importantly, visualizing gene expression changes upon SAHA treatment demonstrated large-scale transcriptional changes, which were consistently observed in new RNA profiles at both treatment durations (Fig. 1g, h). We hypothesized that intronic reads would provide an alternative approach to examining the direct effects of compound

treatments, as introns are primarily found in newly transcribed pre-mRNA, and intronic reads have previously been reported to exhibit similar signals to SLAM seq[16]. However, the analysis of intronic reads identified only a few SAHA-responsive genes at the single-cell level in our NASC-seq2 data (Supplementary Fig. 1f–g).

To gain further mechanistic insights into the transcriptional effects of SAHA, we reanalyzed comprehensive ENCODE ChIP-seq data for 286 transcription factors and regulators profiled in K562 cells to assess whether specific factors were enriched in genes directly responding to SAHA treatment (comparing up- vs. down-regulated genes). No enrichments were found among the significantly responding genes detected at the total RNA level after 30 min. Conversely, when analyzing new RNA profiles, we identified five significant factors, including BRD4 (Supplementary Fig. 2a, b). After 60 min of SAHA treatment, the analysis of total RNA profiles identified only a few significant factors (adjusted *p*-values < 0.05; Wilcoxon test) (Fig. 1i). In contrast, the analysis of new RNA profiles showed a substantial enrichment of many transcription factors in the promoters of SAHA-responding genes (adjusted *p*-values < 0.05; Wilcoxon test) (Fig. 1j and Supplemental Data 2), with the strongest enrichments for factors containing bromodomains, such as BRD4 and its frequent interaction partner YY1. Since SAHA inhibits HDACs from removing acetylation marks from histone tails, and bromodomains bind to acetylated histone tails, our results further support that the changes observed in new RNA reflect the mechanistic functions of SAHA. SAHA's specificity to inhibit deacetylation is not limited to HDACs[17], which may explain why the binding of several other transcription factors is highly enriched among the up- and downregulated genes. The binding of HDAC1 and HDAC2 was enriched among the upregulated genes after SAHA treatment in the new RNA analysis (Fig. 1j). Analysis of histone modification ChIP-seq datasets for untreated K562 cells showed a strong enrichment for H3K27Ac at the upregulated genes, consistent with the known effect of SAHA on H3K27Ac in NK cells[18] (Fig. 1k). We conclude that profiling newly transcribed polyA+ RNA can capture direct transcriptional effects of epigenetic compounds as early as 30- or 60-min post-treatment.

Gene transcription in eukaryotes typically occurs in short bursts, and single-cell RNA-sequencing has become a crucial strategy for inferring the kinetic parameters of endogenous genes in a transcriptome-wide manner[14,19,20]. To explore the mechanisms by which SAHA upregulates genes that are directly responsive, we separately inferred transcriptional bursting kinetics for untreated and SAHA-treated individual cells. Subsequently, we identified genes exhibiting significant changes in burst frequency or size. Interestingly, alterations were observed in both transcriptional burst frequencies and sizes (Fig. 2a, b). Notably, increased burst frequency was most often observed for upregulated genes (69% and 77% at 30 or 60 min, respectively), with significantly more genes affected in terms of frequency at 60 min ($P = 2.1e\text{-}5$, Binomial test) (Fig. 2c). In contrast to the

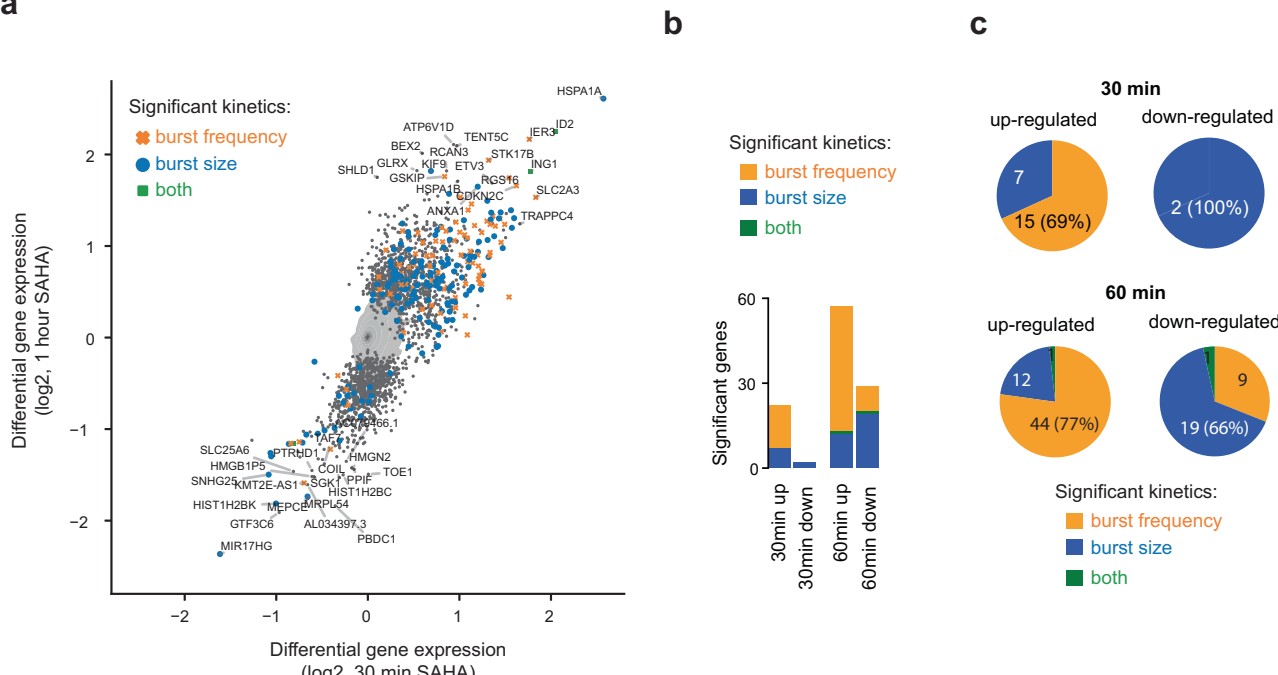

**Fig. 2 | Transcriptional bursting effects of SAHA treatment in K562 cells.**
**a** Scatter plot of differential gene expression after 30 min or 1 h SAHA treatment in K562 cells (new RNA analysis), where genes with significant transcriptional bursting change in either burst size, burst frequency or both are colored coded. **b** Barplot summarizing the numbers of genes with significant changes in transcriptional burst frequency, size or both, stratified by treatment duration and the direction of new RNA change (up- or down-regulated). **c** Pie charts comparing the numbers of genes (and percentages) with certain bursting kinetic changes, for the respective time-points and direction of overall new RNA change. Source data are provided as a Source data file.

upregulated genes, those downregulated after 60 min of SAHA treatment were significantly more likely to show reduced burst sizes rather than burst frequencies ($P = 9 \times 10^{-5}$, chi-square test). At 30 min, however, very few genes were downregulated, suggesting that the direct effect of HDAC inhibition is gene activation through increased burst frequencies.

Having identified direct transcriptional effects of SAHA using NASC-seq2 at single-cell resolution, we sought to adapt the method for large-scale analysis of various epigenetic compounds (Fig. 3a, "**Methods**"). NASC-seq2 suffers from a bottleneck inherent to drug screening on the single-cell level whereby, large-scale screens quickly become costly and time-consuming, since each compound needs to be administered to hundreds to thousands of individual cells. We therefore adapted NASC-seq2 for analyses of smaller cell populations (hereafter called mini-bulks, with 10–100k cells per well), enabling more drugs to be screened in a single experiment. To computationally separate new and pre-existing RNAs, we employed a binomial mixture model[21] for each gene in every treatment condition. We also optimized this model for GPU computing to facilitate rapid computational analysis ("**Methods**"). Simulations indicated reliable inference of new fractions of RNA, with our data's signal-to-noise ratios situated in the stable region of parameter space (Supplementary Fig. 3).

To validate the accuracy of the 4sU-based new RNA profiling in mini-bulks and the computational inference, we treated K562 cells with 4sU in the presence or absence of the transcriptional inhibitor Actinomycin D (ActD), while negative control cells did not receive 4sU. Encouragingly, cells treated with ActD exhibited significantly lower new RNA signals, similar to negative control cells (Fig. 3b), demonstrating that the mini-bulk implementation of the 4sU method reliably captures newly transcribed RNAs.

Next, we conducted a mini-bulk experiment on K562 cells treated with 10 μM 4sU and SAHA for 60 min, which showed concordance among significantly responding genes with the single-cell experiment (Supplementary Fig. 4a, b). The SAHA-responding genes from the mini-bulk experiment could effectively distinguish between individual SAHA-treated and untreated cells (Supplementary Fig. 4c), and the responding genes displayed comparable ChIP-seq enrichment of factors in their promoter regions (Supplementary Fig. 4d). Interestingly, analysis of intronic (but not exonic) reads after SAHA treatment identified responding genes with a similar set of factors enriched in the promoter regions (Supplementary Fig. 4e–g), suggesting that intronic signals are comparable to 4sU-labeled new RNA signals in mini-bulk experiments. The capability to use the intron read signals in mini-bulk samples, unlike the single-cell SAHA experiment, was due to a much-improved overall detection of intronic reads in the mini-bulk experiments (Supplementary Fig. 4h). Possibly, in single cells the 4sU incorporation or alkylation might negatively impact the subsequent reverse transcription, resulting in a 3' end bias (Supplementary Fig. 4h).

Having demonstrated that scalable mini-bulk new RNA profiling can identify direct transcriptional effects, we proceeded to treat K562 cells with an expanded panel of 28 different HDAC inhibitors[22] (Supplementary Fig. 5) at a concentration of 12.5 μM each. For each compound, we identified the significant differentially expressed genes by comparing the new RNA profiles of the treated samples (3 biological replicates) to untreated controls (10 replicates). We applied principal component analysis (PCA) to discern the main patterns of variation in new RNA profiles upon treatment with the different HDAC inhibitors (Fig. 3c). Interestingly, the small molecule inhibitors of HDACs grouped into two categories based on their structural characteristics. Hydroxamic acid HDAC inhibitors elicited similar transcriptional responses, characterized by a shared induction and repression of target genes. Hierarchical clustering of the union of differentially expressed genes following treatment revealed three main clusters of

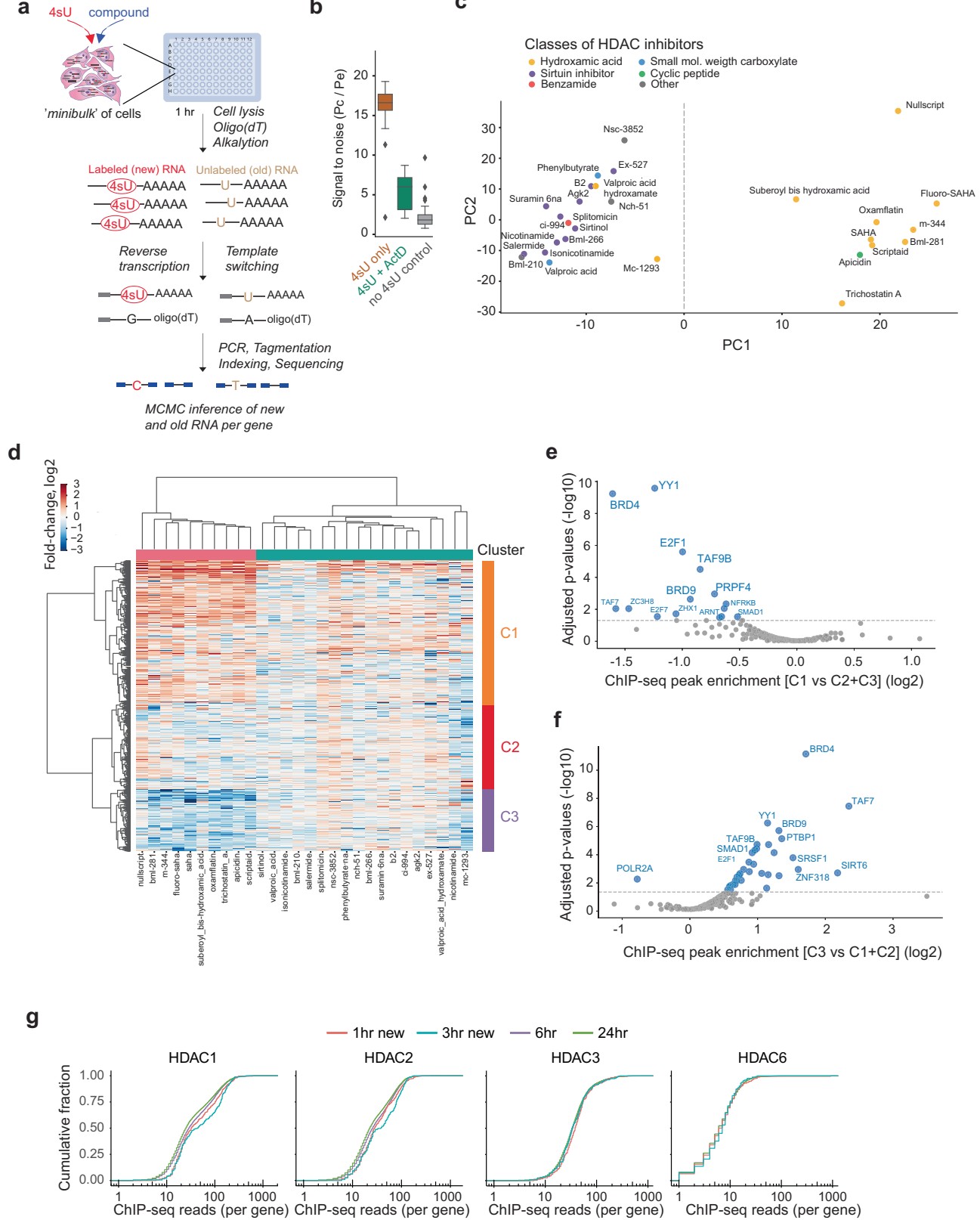

response genes, with the strongest inductions and repressions forming the two largest clusters (Fig. 3d). By integrating these genes with ENCODE ChIP-seq data, we identified specific transcription factors that were enriched in binding to the promoters of the genes separating the HDACi compounds. These factors exhibited opposing effects on the genes in the two clusters. The binding of several factors, notably BRD4,

TAF7, YY1 and BRD9, were depleted in promoters of the genes upregulated after SAHA treatment (cluster C1) (Fig. 3e). Conversely, the promoters of downregulated genes (cluster C3) were enriched for a similar set of factors (Fig. 3f). We propose that this set of factors mediates the direct transcriptional response to hydroxamic acid HDAC inhibitors.

**Fig. 3 | New RNA profiling after perturbing K562 cells with different HDAC inhibitors. a** Illustration of the mini-bulk strategy for large-scale new RNA profiling after chemical compound perturbations. **b** Boxplots showing the signal-to-noise ratios (4sU induced conversions divided by sequencing errors) in experiments where K562 cells were exposed to 4sU alone (2 h; $n = 12$), 4sU combined with the transcriptional inhibitor Actinomycin D (ActD) (2 h, $n = 8$) and untreated negative control cells (no 4sU control, $n = 12$). Boxplots show median, boundaries (first and third quartile), and whiskers denote 1.5 times the interquartile range of the box. **c** Scatter plot showing the two first principal components (PC1 and PC2) for the panel of 28 HDAC inhibitors. For each compound, the mean of each biological replicate is shown, and each compound is colored according to inhibitor class[34]. **d** Heatmap showing the union of genes that were significantly up- or down-regulated at the new RNA level when comparing each HDAC inhibitor against DMSO-treated control cells. The compounds (columns) are ordered by their

PC1 score. For genes (rows), the average difference against DMSO-treated control cells is shown (log2) across three biological replicates per compound. Genes were clustered using hierarchical clustering into four main clusters (C1 – C4). **e, f** Volcano plots showing factors with enriched ChIP-seq binding in promoters of specific gene clusters. In **e**, cluster C1 is compared to the union of the genes in clusters C2 and C3; in **f**, cluster C3 is compared to the union of genes in clusters C1 and C2. **e, f** Enrichments were evaluated using Fisher's exact test (two-sided), with the resulting P values adjusted using the Benjamini-Hochberg procedure. **g** Cumulative distributions of HDAC ChIP-seq reads in promoters of genes identified as significantly differentially expressed after SAHA exposure (5 μM). Comparisons are shown for new RNA analysis after 1 or 3 h of SAHA exposure, and standard total RNA-seq analysis after 6 and 24 h of SAHA treatment. Source data are provided as a Source data file.

We aimed to evaluate the extent to which the direct transcriptional effects detectable in new RNA after 30 or 60 min remain detectable in standard RNA-seq experiment at the total RNA level, even after prolonged SAHA treatments. To accomplish this, we generated RNA-seq libraries from K562 and HEK293 cells treated with SAHA at concentrations of 10 μM and 5 μM for durations of 6 and 24 h, time points which are commonly used in systematic compound screening projects[2–4]. Additionally, we created mini-bulk samples where SAHA was administered at a dose of 12.5 μM for 3 h, with 4sU added only during the final hour before cell harvesting for new RNA profiling. Standard RNA-seq at 6 or 24 h consistently identified thousands of differentially expressed genes in both cell lines (Supplementary Fig. 6a–e), with minor dose-dependent variations. However, within this extensive gene set, identifying those directly responding to SAHA treatment was not feasible (Supplementary Fig. 6e). New RNA profiling revealed direct transcriptional effects that longer treatments failed to detect—or diluted—when analyzing HDAC enzyme binding occupancy at the promoters of the differentially expressed genes. HDAC1 and HDAC2 binding to induced genes was significantly more pronounced in new RNA profiles after 1- and 3-h SAHA treatments (with 1 h 4sU exposure) compared to total RNA profiling at 6 and 24 h ($P < 0.001$, Mann-Whitney $U$ test) (Fig. 3g). This suggests that secondary downstream effects, rather than HDAC-specific effects, start to shape the overall patterns of gene expression when examined after 6 h.

In an effort to extend our study to other classes of compounds affecting key enzymes in transcriptional regulation, we treated K562 cells with 83 distinct compounds that either inhibited or activated different families of epigenetic modifying enzymes, such as sirtuins, acetyltransferases, methyltransferases, demethylases, and bromodomain proteins. Each treatment was administered at a dose of 12.5 μM, accompanied by 4sU for 60 min before cell harvesting, and three biological replicates were performed for each compound. Large-scale analysis of new RNA profiles after 60 min of treatment revealed that bromodomain inhibitors caused the most significant overall down-regulation of genes, while histone demethylase inhibitor treatment led to the greatest increase in transcription (Fig. 4a, Supplementary Fig. 7 and Supplemental Data 2). PCA identified several components capturing new RNA changes shared among multiple epigenetic compounds (Fig. 4b, c). Strikingly, PCA analysis of genes identified by differential intronic read signal after each epigenetic perturbation showed a separation along principal components (i.e., same compounds with similar effect sizes) that closely mirrored the PCA results derived from new RNA profiles (Supplementary Fig. 8). The same subset of compounds (hydroxamic acid HDACs and the bromodomain inhibitors nvs-1, pfi-1, sgc-cbp30, and bromosporine) separated with similar directions and effect sizes in the respective datasets. Visualizing the first four principal components revealed shared new RNA expression patterns shared among bromodomain inhibitors (PC1) and histone deacetylase inhibitors (PC2 and PC4), as well as more complex

patterns not limited to specific to enzyme targets (PC3) (Fig. 4c and Supplementary Fig. 9). Further exploration of the strongest patterns of new RNA changes showed bromodomain inhibitors separating into two main compound groups and four gene clusters (Fig. 4d). Specific DNA-binding or associated factors were enriched in the promoters of genes in clusters 1 (C1) and 4 (C4) (Fig. 4e,f). Notably, down-regulated genes in cluster 4 (C4) were enriched for multiple factors, including YY1, CREM, GAPBP1, TAF1, ATF7, while the genes upregulated by bromodomain inhibitors showed a lack of binding by many of the same factors (e.g. EP400, CREM, YY1). Interestingly, HDAC6 and TAF15 binding was enriched among genes upregulated by bromodomain inhibitor treatment. Analyzing DNA methylase inhibitors (METi) revealed that the patterning of compounds mainly corresponded to varying increases in new RNA (Fig. 4g). Among enriched factors at these promoters, the chromatin remodeling subunits SMARCA4 and SMARCE1 were identified in cluster C2 (Fig. 4h), with overall lower enrichment in cluster C3 (Fig. 4i). Importantly, the compound-specific gene expression responses to treatments observed (Fig. 4a), together with the specific enrichments of ChIP-seq bound factors at the promoters of responding genes, demonstrated that the transcriptional effects of the compounds tested were mechanistically informative rather than generic (e.g., stress responses).

Extending the new RNA profiling paradigm to adherent cells and compounds targeting broader cellular processes, we conducted mini-bulk new RNA profiling of adherent breast cancer MCF7 cells following treatment with a panel of 40 breast-cancer-relevant compounds. For each compound, we treated MCF7 cells at five different concentrations spanning from 1 nM to 10 μM and constructed mini-bulk libraries that were subsequently sequenced. Since we did not expect these compounds to have their primary action mediated through a direct interference of the transcriptional machinery, we treated the MCF7 cells for 4 h prior to new RNA profiling. Together with nine untreated control MCF7 cell preparations, this design allowed us to fit dose-response curves for each gene and drug, with associated p-values estimating drug responsiveness (example in Supplementary Fig. 10a). At least one significant gene was found for 13 of the drugs after multiple testing correction at a 5% false discovery rate (Supplementary Fig. 10b, Supplementary Data 3). Performing hierarchical clustering on the dose-response slopes identified for all drugs across more than 3,000 genes revealed clustering of compounds by drug target for only certain classes, including the PI3K/AKT/mTOR inhibitor class (Fig. 4j). To identify affected pathways and functions, we conducted a gene ontology overrepresentation analysis. This method, while needing more than just a few genes, is robust against random false positives. Consequently, we adopted a higher false discovery rate (50%). Interestingly, the new RNA profiles showed that inhibitors of phosphoinositide 3-kinase and mTOR impacted ribosomes and translation (Fig. 4k). MCF7 cells carry PIK3CA mutations that have been clinically associated with vulnerability to PI3K inhibitors such as alpelisib[23] and

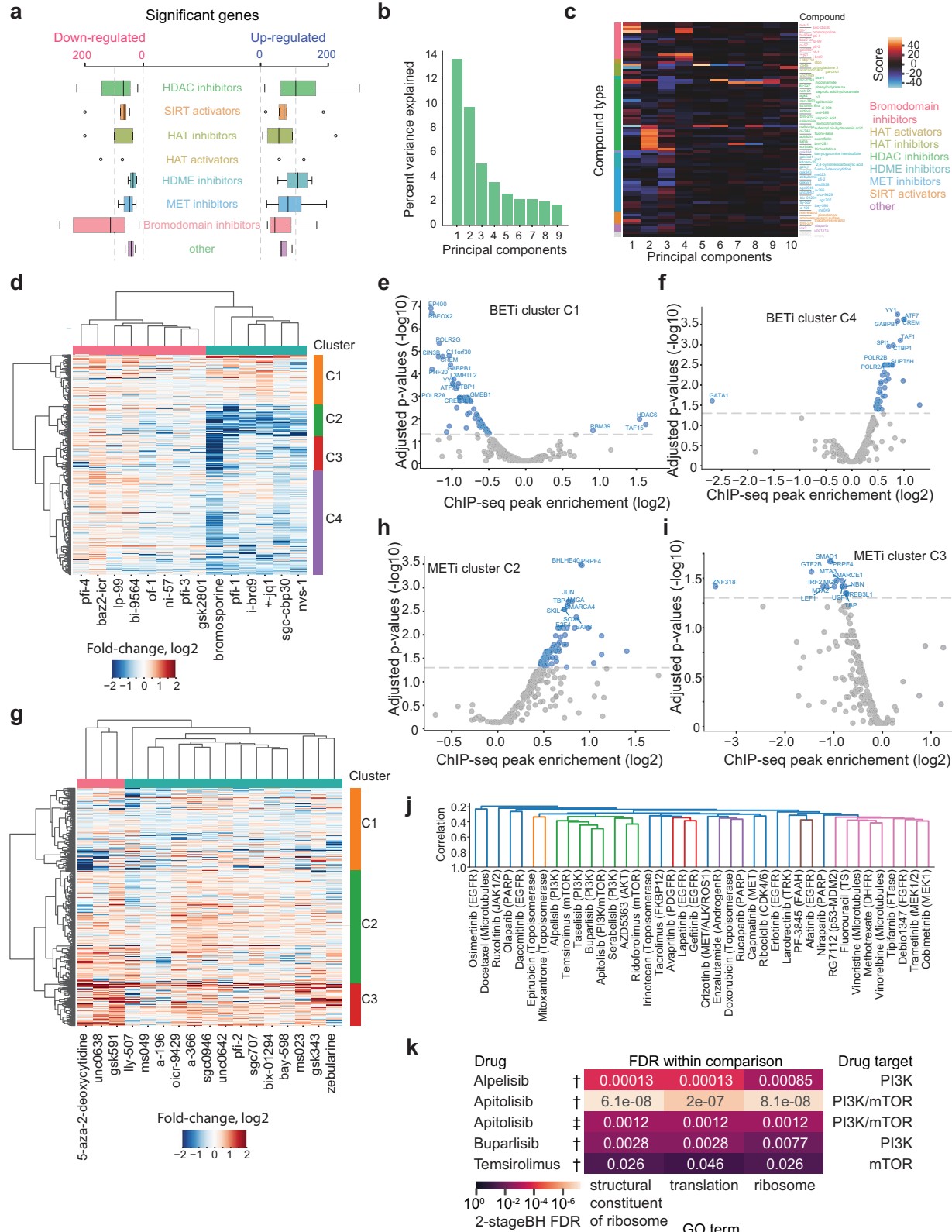

previous studies have shown that inhibition of the PI3K/mTOR pathway leads to translational inhibition through loss of phosphorylation of the 4E-BP translation initiation factor[24,25]. Thus, new RNA profiling corroborated that phosphoinositide 3-kinase and mechanistic target of rapamycin (mTOR) inhibitors impact translation. No other drugs or gene ontology terms remained significant after correcting for multiple testing for gene ontology terms. To prevent bias favoring highly expressed genes, we employed two strategies: comparing down-regulated genes against up-regulated ones, to balance out the bias, and using expression-matched background gene sets. Although doxorubicin, a DNA-damaging drug, significantly reduced the expression of numerous genes (as shown in Supplementary Fig. 10b), there was no significant pattern observed in any gene ontology terms, indicating a non-specific response to DNA damage at our time point of analysis.

**Fig. 4 | Large-scale new RNA analysis of epigenetic compounds effects in K562 cells. a** Boxplots showing the number of significantly up- and down-regulated genes per compound, grouped by mode of action. Boxplots show median, boundaries (first and third quartile), and whiskers denote 1.5 times the interquartile range of the box. Sample-sizes per group (HDAC inhibitors: 28, SIRT activators: 5, HAT inhibitors: 5, HAT activators: 2, HDME inhibitors: 6, MET inhibitors: 17, Bromodomain inhibitors: 14, other: 3). **b** Principal components and the percentage of variance explained by each component for the analysis of all 83 epigenetic compounds. **c** Heatmap showing the scores of each compound onto principal components 1 to 10, where compounds (rows) are colored according to their overall mode of action. **d** Heatmap of average new RNA differences for significantly up- or down-regulated genes, comparing each bromodomain inhibitor against DMSO-treated control cells (three biological replicates). Both compounds (columns) and genes (rows) where subject to hierarchical clustering. **e, f** Volcano plots showing enrichment of transcription factor binding (ChIP-seq) in the promoters of genes in clusters C1 (e) and C4 (f), compared to the union of genes in the other clusters. **g** Heatmap with average new RNA differences for significantly up- or down-regulated genes (as in d) comparing each DNA methylase inhibitor against DMSO-treated control cells. (**h,i**) Volcano plots showing factor binding enrichment in the promoters of genes in clusters C2 (h) and C3 (i), respectively. **e, f, h, i** Enrichments were evaluated using Fisher's exact test (two-sided), with the resulting P values adjusted using the Benjamini-Hochberg procedure. **j** Average linkage hierarchical clustering of breast cancer MCF7 compound responses on fitted dose-response slopes for 3479 genes. Colors indicate correlation strength ($r = 0.33$ cutoff), highlighting the PI3K/AKT/mTOR inhibitor class. Drug targets are shown in parentheses. **k** Significant gene ontology enrichment of compound-responsive genes identified from new RNA profiling of MCF7 cells. Downregulated genes were compared to expression-matched genes or upregulated genes, with P-values corrected for multiple testing within each of the resulting in 128 comparisons for 40 compounds that passed quality-control. Source data are provided as a Source Data file.

## Discussion

In this study, we explored methods to identify the direct transcriptional effects of small molecules that target epigenetic processes. Small molecules can quickly inhibit or stimulate specific protein functions, making them excellent candidates for mapping direct transcriptional effects due to their temporal precision. We demonstrated that 4sU-based new RNA profiling of single cells and mini-bulk cell populations can detect transcriptional changes in hundreds of genes following SAHA treatment for 30 or 60 min. Importantly, the subset of genes showing direct transcriptional changes after SAHA treatment was enriched for binding of specific transcription factors and regulators in their promoters. Upregulated genes exhibited a decrease in BRD4 and YY1 binding and, while YY1 binding was enriched in the promoters of downregulated genes (Fig. 2e, f). The transcriptional effects of YY1 binding are highly context-dependent, and its effect can influenced by its acetylation status, where acetylation turns YY1 into a transcriptional repressor[26]. SAHA acts as a broad HDAC inhibitor affecting multiple HDAC classes, with actions extending beyond histone deacetylation[27]. Consequently, SAHA treatment likely induces global hyperacetylation in both histone and non-histone proteins. It is plausible that SAHA treatment, through its inhibition of HDACs, results in increased acetylation of HDAC targets such as YY1[28], enhancing its repressive action, which could explain the observed downregulation of YY1 target genes.

The ability to profile the transcriptional response of numerous compounds in a single experiment revealed shared patterns of direct transcriptional responses. Among HDAC inhibitors, hydroxamic acid-based compounds triggered the most immediate response in K562 leukemic cells, suggesting that these cells may depend heavily on specific transcriptional programs targeted by these compounds. This level of resolution is crucial for attaining a mechanistic, target-based readout from drug screenings in cell populations—an insight unattainable through standard RNA-seq experiments.

A previous study analyzed the transcriptional effects of Banp removal[16] and found that intronic reads carried similar information to SLAM-seq signals. In this study, we systematically compared analyses of intronic reads with the new RNA signal from 4sU labeling. At the single-cell level (Fig. 1 and S1), intronic reads had limited utility for identifying direct transcriptional effects, possibly due to underrepresentation of intronic reads in single-cell 4sU experiments for unknown reasons. Conversely, at the mini-bulk level, differential expression based on intronic reads (without using 4sU labeling to distinguish newly transcribed RNA) captured similar direct effects as analysis of labeled new RNA expression levels. This finding suggests that re-analyzing earlier RNA-seq data conducted immediately after genetic and chemical perturbations using intronic expression levels might provide new insights into direct transcriptional effects.

Based on the SLAM seq strategy[6], where new RNAs can be computationally separated by specific base conversions, we streamlined and automated the experimental steps and extended the computation with GPU-accelerated inference. In this study, we conducted hundreds of 4sU-based new RNA profiling experiments of mini-bulk cell populations, showcasing the scalability of the method and its applicability across multiple cell lines (both adherent and suspension-grown). By eliminating the bottlenecks in library preparations, future challenges with large-scale perturbation experiments will shift towards logistics, particularly in maintaining temporal precision while treating numerous microwell plates with varied compounds, concentrations and time points. Fast, scalable solutions for new RNA profiling (and potentially intronic analyses of standard RNA-seq) will be broadly beneficial for examining the direct transcriptional effects of small molecule compounds. For instance, we envision future studies mapping direct transcriptional effects of candidate drugs targeting a range of cellular processes to help understand molecular targets and predict off-target effects, although the 4sU labeling times have to be optimized based on when the first transcriptional changes occur. It will be equally exciting to use this new RNA profiling strategy to map the effects of specific transcription factors and regulators (after removal or overexpression) on a larger scale to dissect their direct transcriptional targets and unravel the mechanisms through which they control gene expression.

## Methods
### Cell culture
K562 cells were grown in RPMI 1640 medium (Gibco) supplemented with 10% fetal bovine serum (Sigma) and 1% Penicillin/ Streptomycin (HyClone) at 37 °C and 5% $CO_2$. Cells were grown in T25- flasks, seeded to a density of $2 \times 10^5$ cells/ ml and split 1:4 once the density approached $10^6$ cells/ ml, which usually took place every three to four days.

HEK293FT cells were grown in complete DMEM medium (Gibco) supplemented with 4.5 g/l glucose (Sigma), 6 mM L-glutamine (Gibco), 10% fetal bovine serum (Sigma), 0.1 mM MEM nonessential amino acids (Gibco), 1 mM sodium pyruvate (Gibco) and 100 μg/ml Penicillin-Streptomycin. Cells were grown in T-25 flasks and dissociated with TrypLE Express (Gibco) before re-seeding at 1:3 density every three to four days.

**MCF7 cell culturing.** MCF7 breast cancer cell line cells were seeded at a density of 50,000 cells per well on the in a 96-well plate on the day before the experiment and were allowed to equilibrate and attach to the culture surface overnight. On the morning of the experiment, cells were treated with a panel of 40 FDA-approved breast cancer drugs, resuspended in DMSO in a dilution series to reach final concentrations in the culture medium of 1, 10, 100, 1000 and 10000 nM. At the same time, 4sU at a final concentration of 200 μM was administered and the cells were treated with drug and 4sU for four hours before cell harvest.

## Generation of NASC-seq2 libraries

NASC-seq2 is an improved version of NASC-seq[9], with significantly improved throughput of cells and increased sensitivity. A detailed protocol of NASC-seq2 is available at protocols.io[29]. K562 cells were treated with SAHA at 10 μM concentration and labeled with 4sU at a concentration of 200 μM at either 60 or 30 min before cell harvest, thereafter cells were spun down for 5 min at 300 RCF, resuspended in DPBS (Gibco), stained with propidium iodide (Invitrogen) and sorted into 384 well plates using a FACSMelody cell sorter (BD biosciences) with a 100 μM nozzle into 384-well PCR plates containing 0.3 μl NASC-seq2 lysis buffer containing 2.5 U/μl recombinant RNASe inhibitor (RRI, 40 U/μl, Takara) and 0.1 % Triton X-100, overlaid by 3 μl Vapor-Lock (Qiagen). The plates were kept at −80 °C until prepared for sequencing using NASC−seq2. Briefly, 0.3 μl alkylation mix containing 50 mM Tris-HCl (pH 8.4), 10 mM iodoacetamide (Sigma-Aldrich) and a final concentration of 45% DMSO (Sigma-Aldrich). The samples were alkylated for 15 min before the reaction was quenched at room temperature for 5 min using 0.4 μl quenching mix (35 mM dithiothreitol (DTT, Sigma), 0.5 mM dNTPs, 0.6 μM oligo-dT primer (5′-Biotin-ACGAGCATCAG-CAGCATACGATTTTTTTTTTTTTTTTTTTTTTTTTTTTTTVN-3′; IDT) and 0.4 U/μl RRI). Following the quenching reaction, the cells were denatured at 72 °C for 10 min and 3 μl RT reaction (2 μl TSO (5′-Biotin-AGAG ACAGATTGCGCAATGNNNNNNNNNrGrGrG-3′; IDT), 25 mM Tris-HCl (pH 8.0), 35 mM NaCl, 1 mM GTP (tris-buffered, Thermo Fisher Scientific), 2.5 mM MgCl2, 5 % PEG, 2 mM DTT, 0.4 U/μl RRI, 2 μM TSO and 2 U/μl Maxima H-minus reverse transcriptase (Thermo Fisher Scientific)) was added and the plates were incubated at 42 °C for 90 min followed by 10 cycles of 50 °C and 42 °C for 2 min each, and final denaturation for 5 min at 85 °C. Next, 6 μl preamplification PCR mix (1x KAPA HiFi PCR buffer (2 mM Mg at 1x, Roche), 0.02 U/μl KAPA HiFi HotStart DNA polymerase (Roche), 0.5 μM forward primer (5′-TCGTCGGCAGCGTCAGATGTGTAT AAGAGACAGATTGCGCAATG-3′; IDT) and 0.1 μM reverse primer (5′-AC GAGCATCAGCAGCATACGA-3′; IDT), 0.3 mM dNTPs and 0.5 mM MgCl2) was added to each well and PCR was performed at 98 °C for 3 min, 21 cycles of 98 °C for 20 s, 65 °C for 30 s and 72 °C for 4 min, followed by 5 min final extension at 72 °C.

The amplified cDNA was cleaned up using 6 μl of home-made SPRI beads in 22% PEG and eluted in 12 μl H2O. Cleaned up libraries were quantified using the QuantiFluor dsDNA assay (Promega) on a FLUOstar Omega (BMG Labtech) fluorometer and the cDNA libraries were diluted to 200 pg/μl. 1 μl of the diluted libraries were transferred to a new plate and 1 μl tagmentation buffer (10 mM Tris-HCl (pH 7.5), 5 mM MgCl2 5 % N,N-dimethylformamide and 0.08 μl Tn5 enzyme (ATM, Illumina Nextera XT sample preparation kit) was added. The tagmentation reaction was carried out at 55 °C for 10 min and the reaction was stopped by the addition of 0.5 μl 0.2% SDS. Tagmented libraries were then amplified through the addition of 1.5 μl of custom Nextera index primers (0.5 μM each) and 3 μl tagmentation mix (1x Phusion HF buffer (Thermo Fisher Scientific), 0.2 mM dNTPs and 0.01 U/μl Phusion DNA polymerase (Thermo Fisher Scientific)). The PCR protocol was carried out with 3 min gap filling at 72 °C before initial denaturation at 98 °C for 3 min and 12 cycles of 98 °C for 10 s, 55 °C for 30 s and 72 °C for 30 s, with final elongation of 5 min at 72 °C. Final indexed library was pooled and purified using 0.6x volume home-made SPRI beads in 22% PEG and their concentrations were quantified using the dsDNA Qubit kit (Invitrogen). The resulting cDNA libraries were sequenced on a MGI DNBSEQ-G400 instrument with a sequencing setup of 150-bp paired-end reads and 10bp-index read.

## Compound panel

Compounds were obtained from the compound center at SciLifeLab in Solna, Stockholm. Our compound panel consisted of the SGC bromodomain and Enzo epigenetic compound sets for a total of 82 different compounds and were delivered in 250 nl DMSO at a concentration of 10 mM.

## Treatment of cells with compound library

Before each experiment, 20,000 cells were added in 45 μl of growth medium into a round bottom 96-well culture plate (Corning) and allowed to equilibrate for 60 min and the compound library was diluted to 25 μM in warm growth medium and placed in the incubator to keep warm. At the start of each experiment, 45 μl of the diluted compounds were added to the cells, yielding a final compound concentration of 12,5 μM. For experiments using 60-min treatment times, 4sU was added at to a final concentration of 200 μM concurrent with compound treatment. For experiments using 3-h treatment time, 4sU was left out during the first two hours of compound treatment and added to a final concentration of 200 μM during the last hour before cell harvest.

## Transcriptional blocking and negative control experiment

To begin the experiment, 25,000 HEK293FT cells were seeded into a 96-well cell culture plate (Corning) and allowed to equilibrate in the incubator for 1 h. Cells that were undergoing the transcriptional blocking were treated with Actinomycin D at a concentration of 5 μg/ml and 200 μM 4sU, while the control cells were given neither. After 2 h of incubation, cells were spun down for 5 min at 300 RCF and RNA isolation was performed using the MGIEasy Nucleic Acid Extraction kit, automated on an MGISP-960 Automated Sample Preparation System.

## Generation of RNA-sequencing libraries from long-term SAHA treated cells

Twenty-five thousand HEK or K562 cells were seeded into 6 respective columns of a 96 well plate and left to equilibrate for 1 h. Subsequently, each column (for the respective cell line) was categorized into the following treatments: 24-h treatment time with negative control (just growth medium), 5uM SAHA, 10uM SAHA and 6-h treatment time with negative control, 5uM SAHA, 10uM SAHA. Pre-dilutions of SAHA were performed with corresponding warmed growth medium. During treatment, cells were kept in the incubator before they were spun down for 5 min at 300 RCF at the end of the incubation. Following the respective treatment of cells, an automated RNA extraction was performed using the MGIEasy Nucleic Acid Extraction Kit on a MGISP-960 pipetting robot. The RNA concentration was measured on a fluorometer (AccuBlue® Broad Range RNA Quantitation Kit) and subsequently standardized to 5 ng/μL. Following this, 4 μL of standardized RNA (~ 20 ng of RNA) was taken to generate libraries using an MGISP-960 pipetting robot as described for all other mini-bulk libraries produced in this work. Briefly RNA samples were directly reverse transcribed, amplified with PCR (12 cycles), indexed, and prepared for 100 bp paired-end sequencing on an MGI-DNBSEQ G400.

## Computational pipeline NASC-seq2

The sequenced reads were aligned using zUMIs (2.9.3e) against the human reference genome (hg38), internally using STAR version 2.7.5b and additional STAR parameters:

'--alignSJoverhangMin 1000 --alignSJDBoverhangMin 1 --bamRemoveDuplicatesType UniqueIdentical --outFilterMismatchNoverReadLmax 1, --outFilterMismatchNmax 10 --outFilterMismatchNoverLmax 0.1 --outFilterScoreMinOverLread 0.66 --outFilterMatchNminOverLread 0.66 --outSAMattributes MD --scoreDelOpen −10,000 --scoreInsOpen −10,000 --limitBAMsortRAM 150000000000'.

UMI containing reads were identified by the pattern ATTGCG-CAATG, and transcript assignment was performed with Ensembl gene annotations GRCh38.95.

## Assignment of molecules as new in NASC-seq2 data

UMI-tagged reads were reconstructed and assigned as new or pre-existing. Briefly, paired end reads with the same UMI-tag were grouped together using stitcher.py (https://github.com/AntonJMLarsson/stitcher.py) and the overlapping consensus sequence containing reconstructed molecules were written to a new bam file. The

reconstructed molecules were compared to the reference genome, and positions of T > C (for positively stranded genes) and A > G (for negatively stranded genes) were recorded and saved. The recorded mismatch statistics for each molecule in each cell were used with the expectation-maximization algorithm described by Jürges et al.[18] to calculate the probability of a mismatch arising from 4sU-induced conversions in new reads ($p_c$). Mismatches arising from errors introduced during library preparation ($p_e$) were estimated as the mean of the C > T and G > A mismatch rates. A likelihood-ratio hypothesis test was then carried out on each reconstructed molecule with the null hypothesis $H_0 : p = p_e$ and alternative hypothesis $H_A : p = p_c$ with a binomial likelihood at α = 0.05.

### Processing of RNA-sequencing data from long-term SAHA treated cells

Filtering, mapping, and transcript counting was performed using zUMIs (v2.9.7) with default STAR parameters. Transcripts were annotated using Ensembl annotations (GRCh38.95). Further analysis was performed in RStudio, where samples were filtered to have min. 1,5 M reads per sample and at least 10,000 genes detected (3 samples removed past QC). Differential gene expression was performed between treatment and control samples using a student's $t$-test. K562 CHiP-seq data was integrated for HDAC1,2,3 and 6 to quantify HDAC target proportions within differentially expressed genes (criteria: fold-change > 1.5 or < 0.6, nom. $p$-value < 0.05) in both 4sU-treated and long-term treated samples. ECDF curves of HDAC CHiP-seq reads were generated using ggplot2 stat_ecdf function. $P$-values were computed using pairwise Mann-Whitney U and Kolmogorov-Smirnov tests, testing for differences in medians/central tendency and overall distributions, respectively.

### RNA isolation for cells treated with compound panel

For the mini-bulk experiments with the panel of 88 compound panel (out of which 83 compounds passed our quality control), RNA was isolated using a modified version of the Bio-on-Magnetic-Beads RNA isolation protocol (protocol 8.2, version 1.0)[9] and automated using the Bravo Automated Liquid Handling Platform (Agilent). Briefly, compound-treated and 4sU-labeled cells were spun down at 300 RCF for 5 min. The supernatant was removed and 24 μl lysis buffer (4 M GITC (Sigma), 50 mM Tris HCl pH 7,6, 2% Sarkosyl (Sigma), 20 mM EDTA (invitrogen)) added to the remaining cell pellet, before 32 μl isopropanol was added to the cell lysate and the solution was incubated for 5 min at room temperature before 4 μl silica-coated MagPrep magnetic beads (Sigma) were added. The nucleic acids in the lysate were allowed to bind to the beads for 5 min before the beads were allowed to settle on a magnetic rack and the supernatant removed. The beads were resuspended in 32 μl isopropanol and washed twice with 40 μl 80% ethanol to remove the last of the lysis buffer. The beads were resuspended in 15 μl DNAse I reaction mix (10 mM Tris-HCl pH 7.6, 0.05% Tween 20 (Sigma), 2,5 mM MgCl$_2$ (invitrogen) and 0.5 mM CaCl$_2$ (Sigma), 20 U/ml DNase I (invitrogen), 0.05% recombinant RNase inhibitor (Takara)) and incubated for 30 min at 37 °C to remove genomic DNA from the reaction. After incubation, 60 μl RNA binding buffer (1 M Gu-HCl (Sigma), 0.05% Tween 20 (Sigma), 100% EtOH) was added again to let the RNA bind the beads, before the beads were washed once with 40 μl 80% ethanol. The purified RNA was lastly eluted in 15 μl nuclease-free water.

For K562 and HEK cells treated with SAHA for longer treatment times, as well as the MCF7 cells, an automated RNA extraction was performed using the MGIEasy Nucleic Acid Extraction Kit (on an MGI SP-960 pipetting robot). The RNA concentration was measured on a photospectrometer (AccuBlue® Broad Range RNA Quantitation Kit) and subsequently standardized to 5 ng/μL. Following this, 4 μL of standardized RNA (~20 ng of RNA) was taken to generate libraries using a fully automated, in-house generated protocol (on an MGI SP-960 pipetting robot). Briefly, samples were reverse transcribed and

cDNA amplified (pre-PCR with 12 cycles). After tagmentation, samples were labeled with barcodes for each well during index PCR. The length distribution of the libraries was measured using the Agilent Bioanalyzer system for automated gel electrophoresis.

### Alkylation of RNA and preparation of mini-bulk sequencing libraries

The alkylation and sequencing library preparation follows a slightly modified version of NASC-seq2 protocol[29]. Briefly, 2 μl purified RNA from each sample was transferred to a 384 well plate and 2 μl alkylation mix (50 mM Tris-HCl pH 8.4, 45% DMSO (Sigma), 200 mM iodoacetamide (Sigma)) was added to each well and the plate was incubated at 50 °C for 15 min. To quench the alkylation reaction, 2 μl quenching mix (35 mM DTT (Thermo Scientific), 0.5 mM dNTPs (Thermo Scientific), 0.5 μM Smart-seq3 oligo-dT primer (5′-biotin-ACGAGCATCAGCAG-CATACGA T30VN-3′; IDT), 0.4 U/μl RRI (Takara)) was added and the plate was incubated for 5 min at 21 °C, then 10 min at 72 °C to denature any RNA secondary structures. Reverse transcription master mix (25 mM Tris-HCl pH 8.0, 35 mM NaCl (Sigma), 1 mM GTP (Thermo Scientific), 3.6 mM MgCl$_2$ (invitrogen), 5% PEG (Sigma), 2 mM DTT (Thermo Scientific), 0,4 U/μl RNase Inhibitor (Takara), Smart-seq3 TSO oligo (5′-biotin-AGAGACAGATTGCGCAATGNNNNNNNNNrGrGrG-3′; IDT), 2 U/μl Maxima H Minus Reverse Transcriptase (Thermo Scientific)) was then added at a volume of 6 μl to each well and the plate was incubated in a thermocycler using the same RT protocol as in NASC-seq2. After reverse transcription, 6 μl PCR mix (1X KAPA HiFi buffer containing 2 mM MgCl$_2$ (Roche), 0.02 U/μl KAPA HotStart DNA polymerase (Roche), Smart-seq3 forward PCR primer (5′-TCG TCGGCAGCGTCAGATGTGTATAAGAGACAGATTGCGCAATG-3′; IDT), Smart-seq3 reverse PCR primer (5′-ACGAGCATCAGCAGCATACGA-3′; IDT), 0.3 mM dNTPs (Thermo Scientific), 0.5 μM MgCl$_2$ (invitrogen)) was added to each well and PCR was performed as follows: 3 min at 98 °C for initial denaturation, 17 cycles of 20 s at 98 °C, 30 s at 65 °C and 6 min at 72 °C. Final elongation was performed for 5 min at 72 °C. The PCR reaction was cleaned up using homemade 22% PEG SPRI beads at a ratio of 0.7:1 beads to sample, and 0.2 ng of each sample was tagmented using the Illumina Nextera XT kit with one-fifth of the volumes stated in the manufacturer's recommendation. Custom 10-bp index primers were used at a final concentration of 0.1 μM. The resulting cDNA libraries were sequenced on a MGI DNBSEQ-G400 instrument with a sequencing setup of 100-bp paired-end reads and 10bp-index read.

### Computational pipeline

The sequencing data were processed using the zUMIs[30] pipeline version 2.9.4.d using the hg38 human reference genome, STAR version 2.7.5b and additional STAR parameters '--alignSJoverhangMin 1000 --alignSJDBoverhangMin 1 --outFilterMismatchNmax 100 --outFilterMismatchNoverReadLmax 0.1 --outSAMattributes MD --scoreDelOpen −10,000 --scoreInsOpen −10,000 --clip3pAdapterSeq CTGTCTCTTATACACATCT'. Transcript assignment was performed with Gencode transcript annotations, version 21.

We define the error probability ($p_e$) of seeing a specific mismatch (T-C on positive stranded genes and A−G on negative strand genes) and estimate it as the rate of specific conversions observed on the opposite strand (T−C for negative stranded genes and A-G for positive stranded genes). This eliminates the need of having 4sU negative controls in each experiment. To estimate the probability of a given mismatch being converted as a consequence of 4sU incorporation ($p_c$), we implemented the Expectation-Maximization algorithm described in Jürges et al[18] and obtain $p_c$ and $p_e$ estimates per treatment.

### Estimating the fraction of new RNA per sample

Using the estimates for conversion and error probabilities, we calculated the fraction new RNA per gene in a cell ($π_g$) using a slight

modification of the binomial mixture model described in Jürges et al[21]. Our mixture model assumes that each mismatch is either due to an error with probability $p_e$ or a conversion with probability $p_e + p_c$. The probability of observing $k$ T−C mismatches in a read containing $n$ T-positions can be written as:

$$P\left(k; p_c, p_e, n, \pi_g\right) = \left(\pi_g\right) * B\left(k; n, p_c + p_e\right) + \left(1 - \pi_g\right) * B\left(k; n, p_e\right)$$

(1)

Where $B(k; n, p)$ is the probability mass function.

The parameters $k$, $p_c$, $p_e$ and $n$ are obtained from the sequencing data, and we utilize Bayes formula to solve for $\pi_g$ according to:

$$P\left(\pi_g; p_c, p_e, n, k\right) = \frac{P\left(k; p_c, p_e, n, \pi_g\right) * P(\pi_g)}{P(k)}$$

(2)

Computationally, this can be solved by evaluating the joint log-probability

$$\sum_i \ln(P(k_i; p_c, p_e, n_i, \pi_g))$$

(3)

for each read $i$ mapping to the gene for a certain proposed value of $\pi_g$. Using Monte Carlo inference, we can make repeated proposals for the value of $\pi_g$ and return a posterior distribution of likely $\pi_g$ values.

The Monte Carlo inference kernel was implemented using TensorFlow Probability[31] and adapted to run on GPUs for faster inference. We used a Hamiltonian Monte Carlo (HMC) engine with 1000 burn-in steps and 5000 iterations (using 10 leapfrog steps with a step size of 0.05 and a target accept probability of 0.65) with a uniform prior probability distribution and an initial guess for $\pi_g$ of 0.1. Using these settings, we saw convergent behavior on simulated data, where the accuracy of the estimation was dependent on the number of reads (Supplementary Fig. 3). We ran one chain of the HMC sampler and returned the mean of the posterior distribution as the point estimate for the $\pi_g$ value, along with the 95% credible interval of the estimate. If a gene had more than 5000 reads mapped in a condition, the reads were downsampled to 5000. If no conversions were detected in any reads mapping to a gene in a condition, the $\pi_g$ value for that gene in that condition was reported as zero.

## Sample QC
**Samples from K562 cells.** To keep a condition for further analysis, we required that the sample had at least 200,000 total reads mapping to it, that 45% of mapped reads were mapping to exonic reads and that the signal-to-noise ratio (defined as $p_c/p_e$) was higher than 10. The impact of this quality control led to that only 83 out of the 88 compounds tested has sufficient samples for differential testing at new and total RNA levels. Median transcript integrity (TIN) values[32] in Supplementary Fig. 4h were calculated across genes. <u>Samples from MCF7 cells:</u> Remove all samples with percent exon less than 45%, total read count less than one million, confidence width of 0.4 or greater, and if any drug-related samples were less than three.

## Differential expression testing for mini-bulk samples
To test for differentially expressed genes, we used a t-test between a gene's expression in the compound-treated samples compared to 10 randomly sampled DMSO controls. Only compounds with three replicates were used, and the DMSO control samples were independently sampled for each test. To account for the low number of replicates, we utilized a variance adjustment approach, in which we first estimated a gene's expected variance based on its expression level

through either linear or lowess regression. If a gene had a lower variance than the regression predicted, we used the predicted value in the t-test instead. After significance testing, we adjusted the p-values for multiple hypothesis testing using two-stage FDR correction.

## Differential expression testing for single-cell samples
Differential expression was performed using DESeq2[7], comparing SAHA treated samples for each time point against treatment time matched DMSO controls.

## Inference of transcriptional bursting from new RNA profiling of individual cells
We applied an existing method to infer transcriptional kinetics from 4sU labeled single-cell transcriptomes[5], adapting it to our dataset. The dataset in this manuscript had too few cells to infer $k_{off}$ from the data, therefore we used the typical off-rate of 100/h typically observed in mouse fibroblasts. We opted to fix $k_{off}$ instead of $k_{syn}$ since $k_{off}$ was recently shown to less variable transcriptome-wide[5] whereas $k_{syn}$ correlated with burst size. We matched the t value to the length of 4sU labeling time periods ($t = 0.5$ h or $t = 1$ h) in our experiment and $k_{deg} = 0.065$/h by calculating $-\ln(1-\text{fraction\_new\_RNA})$ on the data. We calculated burst frequency as $1/(1/k_{on} + 1/k_{off})$ and burst size as $k_{syn}/k_{off}$ and performed pairwise comparisons using a permutation test on the estimates produced prior to the slow maximum likelihood step of the method[5]. P-values from the permutation test undergo a Benjamini-Hochberg correction, using a 5% false discovery threshold. We further refined burst frequency and burst size using maximum likelihood and only kept genes where the direction of change remained the same. Code used for inference is available on Github.

## Principal component analyses
Both the heatmaps and the conventional 2D scatterplots show the same quantities: the score per sample for the principal component in question. For the sake of simplifying visualization, the samples in the heatmaps were initially sorted by similarity using average-linkage Euclidean hierarchical clustering before being reordered by drug class. Principal component decomposition was performed on standardized expression data after dropping genes with no expression information using packages from scikit-learn version 1.2.2 with exact full singular value decomposition.

## Enrichment of DNA-binding factors in promoters
We downloaded ChIP-seq DNA sites (peaks) for transcription factors and cofactors in K562 cells generated via the ENCODE project (https://www.encodeproject.org), selecting experiments without treatments or genetic modifications. If there were multiple data sources for a ChIP target, we chose the data set with the least audit non-compliance warnings and if two datasets had the same number of warnings, we picked the experiment with the highest read depth. This resulted in a set of 286 transcription factors for which we had genome-wide peak data. For each ChIP-seq target and each gene in the Ensembl release 95 assembly GRCh38 annotation, we recorded whether there was a peak within a region consisting of 1000 bp upstream of the transcript start site (TSS) and 100 bp downstream of the TSS, where we expected the promoter region to be. For each ChIP-seq target, we compared how many of the upregulated genes (significant from DE-testing) had peaks in promoters to how many of the downregulated genes had one, obtaining a P-value through Fisher's exact test and a fold change for effect size estimate by dividing the fraction of peak containing genes in the upregulated group by the fraction of peak containing genes in the downregulated genes. The total number of factors tested per comparison depends on how many of its target genes are expressed in any of the two gene sets being compared. P-values were adjusted using the Benjamini-Hochberg method, and a 5% cutoff was applied.

## MCF7 compound panel

Gene ontology tests were based on go.obo from https://geneontology.org/docs/download-ontology/ 2021-03-04 excluding terms defined as obsolete. Expression matching was done by ranking all genes by average expression (lognormmean in Supplementary Data 3), and iteratively go through the list of input genes and search in alternate directions until either a maximum rank difference of 100 or 10% of the genes can no longer have matches added due to having to many nearby genes in the input set. Chi-square test was used if all expected gene counts are above ten, otherwise Fischer's exact test was used. Two-stage Benjamini-Hochberg correction for multiple testing was applied across all tested GO terms.

## Reporting summary

Further information on research design is available in the Nature Portfolio Reporting Summary linked to this article.

## Data availability

Raw sequence data for all experiments (4sU single-cell, 4sU mini-bulk, standard RNA-seq) has been uploaded to ArrayExpress with the accession numbers: E-MTAB-13091, E-MTAB-14822 and E-MTAB-14924. Source data are provided with this paper.

## Code availability

Code for processing and data analyses can be found at GitHub: https://github.com/sandberg-lab/compound-screen (release v1.0 also found at Zenodo[33]).

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

## Acknowledgments

This work was supported by grants from the Swedish Research Council (R.S), the Knut and Alice Wallenberg Foundation (R.S.), the Swedish Cancer Society (R.S.), Karolinska Institutet (R.S.), the Göran Gustafsson Foundation (R.S.), and the Torsten Söderberg Foundation (R.S.).

## Author contributions

R.S., G.-J.H., and L.H. conceived the overall study. G.-J.H. and L.H. developed the mini-bulk 4sU based method. L.H., P.J., and G.-J.H. performed the 4sU experiments of the study (single-cell and bulk). P.J. and

S.H. performed additional RNA-seq experiments. L.H., D.R., G.H., A.L., C.Z., performed computational analyses of the data. L.H. developed the GPU implementation of the new RNA assignment. R.M. and J.H. prepared breast cancer compound panel. R.S. wrote the manuscript, together with contributions from L.H. and D.R.

## Funding

## Competing interests

The authors declare no competing interests.
