## [Transparent Peer Review file · Nature Communications]

Deciphering direct transcriptional effects of epigenetic compounds through large-scale new RNA profiling

Corresponding Author: Professor Rickard Sandberg

Version 0:

Reviewer comments:

Reviewer #1

(Remarks to the Author)

This short manuscript describes data prepared with a metabolic-labeling based RNA-seq strategy to test the transcriptomic effects of a large number of compounds that perturb epigenetic regulators.

The strategy is an update of the authors' previously published protocol NASC-seq, which identifies newly transcribed RNA based on 4sU incorporation, chemical conversion, and detection by mutations, as a number of other, very similar protocols are doing (e.g. SLAM-seq).

One dataset seems to have been prepared at single cell resolution and analysed in this way, while the rest are bulk data made from moderate numbers of pooled cells.

The description of this work as 'development of a method' seems to be a bit of an overstatement; the sample preparation protocol is a modification of existing ones, the analysis approaches are standard, and the employment of automation at some steps is practised commonly. Other than the RNA-seq data itself, no new experimental data is presented.

The data appears to be of high quality though and might have value as a resource for other researchers, although the authors curiously do not describe it as such and do not appear to have deposited it in a publicly accessible repository, and the lack of accompanying chip-seq data for histone marks or epigenetic regulators furthermore limits the data's utility.

Further specific points, mostly relatively minor (except the first):

- Fig. 1i/j (and others): The chip-seq factor analysis seems somewhat uninformative, given that the data is from untreated cells; it would be a lot better to do an actual chip-seq for histone marks and/or other factors upon treatments and compare to untreated cells to check how histone acetylation etc and factor binding is perturbed and how this affects transcription. Is there a reason that histone mark data was not included in the encode data chip-seq analysis?

it would also help to know the total number of factors considered in the chip-seq analysis to put the numbers of significant ones in context.

- The methods do not seem to explain how the bursting parameters were obtained (fig. 2A)

- Some of the text refers to 83 compounds, elsewhere it is 88 compounds – why this discrepancy?

- the data presented in fig. 3 is described as a separate experiment (12.5 microM HDAC inhibitors, 60 min 4sU) when it could very well be just a subset of that presented in fig. 4 (12.5 microM HDAC inhibitors + all other inhibitors, 60 min 4sU). It would be clearer to describe it as a single experiment, eliminating some text, unifying some figures, and focusing on different parts (e.g. HDAC, BET, etc, like it is now) within that context

- Fig. 4c/S8: not entirely clear what was done here (PCA is not mentioned in the methods); the legend for S8 says this is a hierarchical clustering of 'loadings' (why not show the dendrogram?), so I assume the same was done for fig. 4c. If these PCAs were done like the ones shown in fig.3c/S9, the 'loadings' would correspond to the values each gene contributes to

the PCs. However, Fig. 4c/S8 suggests response and input variables have been swapped. Please clarify. The main text further states that 'Strikingly, PCA analyses of genes identified by differential intronic read signal after each epigenetic perturbation resulted in similar principal components as the analyses of new RNA'. It is not clear what precisely is meant by 'similar principal components' or how fig. s8 demonstrates this. Furthermore, the compound labels in fig. 4c are too small.

- Fig. 1e is interpreted as a demonstration that total RNA masks the SAHA effects. An alternative interpretation would be that the new RNA does not capture single cell variation, although I grant that that is unlikely and the observation rather reflective of sensitivity issues. But perhaps it is still worth discussing? Also, is it worth discussing the 'new RNA' plot in fig. 1e as a justification for doing bulk rna-seq, given that the treatments lead to fairly homogenous subpopulations?

- p6, first paragraph: how were the binomial and chi square tests carried out? What exactly was tested, what were H0 etc? same for Anderson-darling on p8 (why wasn't a Mann-Whitney used?)

- p19: it is stated that the prior for π_g is 0.1 – shouldn't this be a probability distribution rather than a fixed value if MCMC sampling is used?

- Fig. S1: Shouldn't the number of newly expressed genes be close to zero without 4sU if the inference is working correctly?

Reviewer #2

(Remarks to the Author)

The manuscript by Hartmanis et al. describes an updated workflow (termed NASC-seq2) for SLAM-seq based nascent mRNA profiling for characterizing cellular responses to drug perturbations. The major advance lies in the implementation of Smart-seq3-like nanoliter liquid handling technology in a 384-well format, which substantially increases the scalability of the assay, e.g. for compound screening applications. The authors first use NASC-seq2 to measure direct effects of SAHA/vorinostat-mediated HDAC inhibition on transcriptome-wide mRNA synthesis in single K562 cells on a scale of hundreds of cells per condition. The authors next apply NASC-seq2 to "mini-bulk" K562 populations (10k-100k cells/condition) treated with an expanded panel of 28 compounds annotated as HDAC inhibitors. This experiment resolved some structure-activity-relationships within the compound panel. Finally, the authors tested 55 more epigenetic-focused compounds (+ the 28 HDACi) and could again stratify compound classes based on acute transcriptional phenotypes with moderate success.

Overall, the manuscript is clearly structured and well written, and the provided methods (including a detailed protocol on protocols.io) are of sufficient detail to implement NASC-seq2. The presented analyses are technically sound and sufficiently support the presented conclusions. The finding that intron-based analyses from total RNA provides similar information content as SLAM-seq based profiling – at least on a 'mini-bulk' level – is interesting and broadly relevant. Nevertheless, NASC-seq2 could serve as a useful tool for medium-throughput screening endeavors with transcriptional readout at high kinetic resolution. However, besides describing a technical advance that increases the scalability of nascent mRNA profiling, the manuscript lacks conceptual and technical novelty and provides very little mechanistic insight. Many of the presented conclusions (e.g. concerning differences between total and nascent mRNA profiling) are well-described in the published literature. Moreover, SLAM-/NASC-seq profiling has been previously applied in low-input settings by the authors and others, and the presented profiling of 83 compounds in K562 cells does not fully demonstrate the utility of NASC-seq2 at scales that are relevant for compound screening.

The following are my major concerns:

1. Both conceptually and technically, NASC-seq2 is largely based on ideas, innovative chemistry, and protocols that were originally described as SLAM-seq, which is only referred to once as "SLAM-seq chemistry". At several points (including the abstract), the authors do not properly distinguish between prior art and technical advances provided by their manuscript. Similar is true for conceptual findings and conclusions. For example, Fig. 1 mainly describes differences between nascent and total mRNA-seq that (i) have been well-described using SLAM-seq and other nascent RNA-seq tools, and (ii) are unsurprising given the short sampling time points in case of total RNA-seq. In general, the authors should give proper credit to previously established concepts and methods and more clearly describe the specific improvements of these methods offered by their manuscript.

2. While NASC-seq2 could provide an important step towards the use of nascent RNAseq for compound screening, the manuscript does not fully establish the scalability and broad applicability of this approach. All presented 'mini-bulk' assays were performed using suspension cells (K562), which can be cultured at much higher densities in multi-well formats compared to adherent cells, which are the more relevant system for compound screens. The authors should demonstrate the feasibility of their 384-well assay in adherent cells. In addition, 83 compounds seems a rather limited number for screening applications overall, and at this scale, more interesting compounds could have been selected to demonstrate relevant 'use-case-scenarios' of their assay. For example, NASC-seq2 could be used to 'de-orphanize' the mechanism of action of novel compounds, to identify and characterize off-target activities, or to benchmark on-target potency. None of this is convincingly shown.

3. The manuscript provides little mechanistic insight into chromatin regulators or the investigated compounds. The presented ENCODE ChIP-seq analyses remain correlative at best and do not yield conclusive insights. Compound selection is

questionable at times, with multiple non-selective, broad-spectrum inhibitors included (e.g., bromosporine, nicotinamide, phenylbutyrate), while more selective chemical probes could have been used to study sub-class specific effects (e.g., Merck60, Entinostat; see e.g., PMIDs 28228643; 35484434), or clinical-stage compounds could have been included to add relevance (e.g., panobinostat, entinostat, pelabresib/CPI-0610, apabetalone/RVX-208). The moderately successful stratification of compound classes in Figures 3 and 4 is likely due to the below-par compound selection, rather than NASC-seq2 technology as such. Fewer but better chemical probes per epigenetic class – perhaps in combination with degron-engineered cell lines – would likely be required for deeper mechanistic insights. Conversely, applying NASC-seq2 to different cell lines or even (patient?) PBMCs exposed to a panel of clinical-grade compounds would add real-world relevance and leverage the single-cell aspect of the technology.

Minor concerns:

- The 'up-regulated' axis in Figure 4a (right half) seems stretched (also the outlier dots seem stretched in the SIRT-activator and HAT-inhibitor boxplots). Please double-check that there was no mishap during figure preparation.
- The axis labels, especially for the ENCODE ChIP-seq volcano plots and all log₂ fold-changes, do not clearly convey what is being compared to what. Please consistently indicate which groups were compared (e.g. 'ChIP-seq peak enrichment log₂[C4 vs. C1-3 transcripts]', ...).
- The value of clustering into exactly 3-4 clusters (Figures 3 and 4) is unclear. Why perform ENCODE ChIP-seq analysis exactly for C2 and C4 in Figure 3d? If analyses are primarily focused on strongest up- vs. down-regulated genes, why not cluster into fewer clusters (e.g., merging C1/C2) or add a log₂ fold-change cutoff averaged on the strongest compounds (i.e. the hydroxamic acid derivatives)
- Were spike-ins used to control for potential global changes in RNA synthesis and if so could you please comment on why you did not deem it necessary in this case?
- There seems to be a typo in the main text linked to Fig. 3e. The text says "Conversely, binding of the same factors was found depleted in promoters of the genes that were downregulated after SAHA treatment (Figure 3e)." We believe panel 3e focuses on Cluster C2, which is the genes mostly upregulated after SAHA treatment?
- The panel of compounds labeled as "BET inhibitors" would more accurately be described as a panel of "Bromodomain inhibitors", as e.g. SGC-CBP30 is an inhibitor of the CBP/p300 bromodomain with ~100-fold selectivity over BET protein (BRD2/3/4/T) bromodomains. This should also be changed in the main text (page 8-9)

Reviewer #3

(Remarks to the Author)

Summary: In this manuscript, the authors developed an updated version of NASC-seq (NASC-seq2) to profile newly transcribed RNA from single cells or mini-bulk samples. They applied this method to measure the direct transcriptional effects of inhibitors targeting epigenetic regulators. The present manuscript is potentially important because the proposed method can be a building block of a platform for large-scale chemical screening quantifying direct transcriptional effects. However, there are several important points that should be addressed to substantiate the authors' claims.

Major points:

1. In Figure 1e, the single-cell transcriptomes appear to be largely affected by the 4sU labeling times, potentially contributing to batch effects. The authors should provide at least two replicates for each condition, and examine this issue carefully.
2. In Supplementary Figure 1d and e, as discussed by the authors, the intronic reads seem to be underrepresented in single-cell NASC-seq2 data and have a limited statistical power to identify differentially expressed genes in terms of new RNA. However, this was not observed in mini-bulk NASC-seq2 data (Supplementary Figure 4g). The authors should carefully examine this issue by investigating the potential causes of this issue (e.g. 3' bias of read coverage within a transcript, low sensitivity).
3. In Figure 3, it would be more informative if the authors provide a list of genes showing primary or secondary downstream effects, and discuss the biological differences of the two sets of genes.
4. In Figure 4, the authors hypothesized that the enrichment of YY1 binding in the promoters of downregulated genes can be explained by the rapid acetylation of YY1 after SAHA treatment. This prediction should be experimentally validated.
5. The authors should discuss the key features of NASC-seq2 by comparing it with NASC-seq.

Minor points:

1. "signal-to noise ratios" should be "signal-to-noise ratios" at page 6.
2. Supplementary Figure 6f referred to at page 8 does not exist.
3. "H_0: p=p_c" at page 14 should be "H_0: p=p_e".
4. At page 18, "algorithm described in Jürges et al11" should be "algorithm described in Jürges et al18".

Version 1:

Reviewer comments:

Reviewer #1

(Remarks to the Author)

The authors have made a good effort to improve the manuscript in response to my comments, although some of it seems to have been done in a hasty way and would benefit from further efforts, as explained below.

- Text explaining the bursting inference has been added but needs proofreading and correction (e.g. '[...] need its inverse since know the count distribution [...], but kon or ksyn'). A ref to the authors' previous paper (Ramskoeld et al, Nat Cell Biol '24) on this has also been added, but it should be noted that bursting inference on 4sU single cell data was to my knowledge first described by Edwards DM et al, Genome Biol, '23, so I'd appreciate if this ref could also be added (also since it is not in the Ramskoeld paper either)

- according to the rebuttal, the bursting inference section is also supposed to contain information on the statistical tests, but it does not. In particular, it is still not clear to me why a KS test (was Anderson Darling in the original manuscript) was used instead of e.g. a Mann-Whitney, when the authors seem interested in showing higher HDAC binding, i.e., a shift in mean or median – a KS/AD could show significance due to different shapes of the distributions even when the median is equal, I think. This is a minor issue, though; I believe the authors' claim of higher HDAC binding.

Further minor issues:

- the compound labels in fig. 4c are still tiny, but perhaps this is ok if readers can zoom in on a pdf

- my PCA queries have been resolved, but the claimed similarities among the new RNA PCA and intron PCA are hard to see, given that fig. 3c only shows HDAC inhibitors and fig S9 all (and with similar blue/green/cyan shades for different groups), and the heatmaps (fig4c, fig.s8) don't appear similar either, apart from the hdac inhibitor 'block' high in PC1 and PC2 in fig. s8 and fig. 4c, respectively.

DH

(Remarks on code availability)

Reviewer #2

(Remarks to the Author)

In their revised manuscript, Hartmanis et al. have comprehensively addressed all my questions and concerns through text editing and extensive additional experiments. I appreciate that conducting an additional screen of 58 compounds in MCF7 cells has been a major endeavor, and I would like to thank the authors for this effort. Together with the original screen in K562 cells, this screen in an adherent cell line, which includes a well-selected panel of compounds, provides a valuable pilot and benchmark for future applications of this technology. I have a few remaining comments and suggestions regarding the presentation of these data:

1. Several of the investigated compounds (e.g. MEKi and others) must be expected to trigger clear transcriptional responses in MCF7 cells after 4 hrs. In this light, the reported effects are surprisingly limited. The authors state that they identified at least one significant gene for 13 of the drugs-For the two MEKi they only identify 2 response genes for Trametinib and none for Cobimetinib, which is far below the expected effect size. The authors should comment on this. I fully agree that this problem might reflect overly stringent statistical tests, and the presented relaxed analysis indeed yields effects sizes in the expected range.

2. To gain confidence in the assay and data, the authors should analyze and present whether functionally related compounds (i.e. inhibitors that hit the same target) lead to similar responses. The presented GO term analysis does not really address this point, and it would be most informative to perform hierarchical clustering or PCA analysis over the entire dataset (under relaxed statistical stringency).

3. While the presented slope analysis over different doses is intriguing, this is only one of many ways to analyze the data. Alternatively, the authors could perform analyses (e.g. drug comparisons described above) with just one of the higher doses. Importantly, to enable readers and potential users to perform such analyses, the authors should provide all primary data of both screens (i.e. primary NGS read counts) as Supplemental Information.

Once more, I would like thank the authors for an excellent revision.

(Remarks on code availability)

Reviewer #3

(Remarks to the Author)

The authors have satisfactorily addressed my comments.

(Remarks on code availability)

Responses to reviewers comments.

Reviewer #1:

This short manuscript describes data prepared with a metabolic-labeling based RNA-seq strategy to test the transcriptomic effects of a large number of compounds that perturb epigenetic regulators.

The strategy is an update of the authors' previously published protocol NASC-seq, which identifies newly transcribed RNA based on 4sU incorporation, chemical conversion, and detection by mutations, as a number of other, very similar protocols are doing (e.g. SLAM-seq).

One dataset seems to have been prepared at single cell resolution and analysed in this way, while the rest are bulk data made from moderate numbers of pooled cells.

The description of this work as 'development of a method' seems to be a bit of an overstatement; the sample preparation protocol is a modification of existing ones, the analysis approaches are standard, and the employment of automation at some steps is practised commonly. Other than the RNA-seq data itself, no new experimental data is presented.

The data appears to be of high quality though and might have value as a resource for other researchers, although the authors curiously do not describe it as such and do not appear to have deposited it in a publicly accessible repository, and the lack of accompanying chip-seq data for histone marks or epigenetic regulators furthermore limits the data's utility.

The reviewer highlights several points that have been addressed in the revised manuscript. Although no mini-bulk procedure exists for 4sU-based RNA-seq, we agree with the reviewer that we primarily adapted an existing procedure to a different format. As such, we have reworded the text to better reflect this. Sequencing data had been deposited in EBI's ArrayExpress (E-MTAB-13091, E-MTAB-14822, accession id pending for additional MCF7 data), and we concur with the reviewer that this represents an important resource. Additionally, it provides conclusive evidence demonstrating that intronic reads in standard RNA-seq capture the same information as using 4sU (30 or 60 min). We believe this is a significant statement that will greatly simplify experimental procedures for capturing direct transcriptional effects.

Further specific points, mostly relatively minor (except the first):

1. Fig. 1i/j (and others)

The chip-seq factor analysis seems somewhat uninformative, given that the data is from untreated cells; it would be a lot better to do an actual chip-seq for histone marks and/or other factors upon treatments and compare to untreated cells to check how histone acetylation etc. and factor binding is perturbed and how this affects transcription.

Is there a reason that histone mark data was not included in the encode data chip-seq analysis? It would also help to know the total number of factors considered in the chip-seq analysis to put the numbers of significant ones in context.

The reviewer raises several questions regarding the impact of epigenetic treatments on histone marks, investigate how those marks relate to the new RNA changes detected, and clarifying the extent of factors used in ChIP-seq analyses.

Starting with the last question, 286 different transcription factors and regulators for which CHIP-seq was available in K562 cells were considered in the previous analyses – thus a comprehensive set of factors. However, the number of factors evaluated in each comparison varies, as a factor is only considered for analysis if any of its target genes are present in either of the two gene sets being compared (for example, SAHA upregulated versus downregulated). In the revised manuscript, we have revised the methods section under the heading "Enrichment of DNA-binding factors in promoters" (pages 22-23) to clarify this point, and we have added the information on the breadth of factors tested to revised manuscript result section (page 5, second paragraph).

To address the comment on the effect of treatment on histone marks, we first note that in the literature it has been reported that SAHA treatment has a strong effect on H3K27 acetylation. For example, SAHA (vorinostat) treatment of NK cells led to a significant upregulation of H3K27Ac marks, consistent with the mechanisms of histone deacetylase inhibition by SAHA (see **Reviewers Figure 1**, pasted below). We have included this reference and added a sentence linking our results to this relevant literature on page 6, first paragraph (line 4) of the revised manuscript.

[REDACTED]

Reviewers Figure 1. Figure 1f-g from Lee et al 2023 (reference 18 in revised manuscript): Showing that SAHA-treatment has a strong positive effect on H3K27Ac marks.

To extend our analysis to impact on histone marks, we reanalyzed our single-cell new RNA expression data against ENCODE CHIP-seq data for histone marks to assess whether differentially expressed genes exhibit enrichment for certain histone modifications. Among the genes upregulated following SAHA treatment (detected at new RNA level), we discovered a significant enrichment ($P < 10^{-8}$; Fisher's exact test) of H3K27Ac in regulatory regions upstream of the transcription start site in untreated K562 cells. Genes expressed, and with high H3K27Ac levels in these upstream regions, became further upregulated due to SAHA-mediated HDAC inhibition (**Reviewer Figure 2**; pasted below; **Figure 1k** in the revised manuscript).

Reviewers Figure 2 (revised manuscript **Figure 1k**). Significant enrichment of specific histone marks in the proximity of SAHA-responsive genes.

Our results from both the histone marks and transcription factor enrichment analyses demonstrate that genes with upstream H3K27Ac are preferentially upregulated upon SAHA treatment. This finding is evident in the enrichment of H3K27Ac marks (new Figure 1k) and in the enrichment of histone acetylation-binding bromodomain proteins (Figure 1i,j). This underscores that analyzing new RNA profiles can identify direct effects of SAHA on the transcriptome, which primarily occur through an increase in the cellular acetylation state.

Altogether, the new analyses suggested by the reviewer have significantly improved our study, as we have strengthened the connection between how treatment with epigenetic compounds impact specific marks on histone tails and, importantly how that correspond to direct transcriptional effects detected through new RNA profiling.

The methods do not seem to explain how the bursting parameters were obtained (**fig. 2A**)

The reviewer correctly noted that the method for inferring bursting parameters was not included in the initial version of this manuscript. We have now addressed this by adding a section titled "Inference of Transcriptional Bursting from New RNA Profiling of Individual Cells" on page 22 of the methods section in the revised version. The development of an accurate computational inference strategy for transcriptional bursting kinetics from new RNA profiles was quite more challenging than inferring kinetics from steady-state RNA profiles, and it is the subject of another manuscript from our lab that is properly referenced in the revised manuscript (reference 5 in the revised manuscript: Ramsköld et al. 2024 Nature Cell Biology). Thus, the revised manuscript includes the methods section for kinetics inference, and we refer the full details of that inference to the referenced study.

Some of the text refers to 83 compounds, elsewhere it is 88 compounds – why this discrepancy?

Thanks for pointing out this inconsistency. The entire set of compounds that we obtained for the experiments presented in the manuscript was 88. However, 5 compounds did not pass the quality control criterias and was therefore removed, resulting in a data set with 83 compounds for analysis. In the revised manuscript, we have updated the text of the revised manuscript to properly refer to pre- and post-quality control compound numbers.

The data presented in fig. 3 is described as a separate experiment (12.5 microM HDAC inhibitors, 60 min 4sU) when it could very well be just a subset of that presented in fig. 4 (12.5 microM HDAC inhibitors + all other inhibitors, 60 min 4sU). It would be clearer to describe it as a single experiment, eliminating some text, unifying some figures, and focusing on different parts (e.g. HDAC, BET, etc, like it is now) within that context.

The reviewer correctly notes that all compound treatments were conducted in a single experiment. Having considered to describe all compounds as one experiment, we maintain that the manuscript is more straightforward to understand when the data is presented in a sequential manner and would prefer to keep this description.

Fig. 4c/S8: not entirely clear what was done here (PCA is not mentioned in the methods); the legend for **S8** says this is a hierarchical clustering of 'loadings' (why not show the dendrogram?), so I assume the same was done for **fig. 4c**. If these PCAs were done like the ones shown in **fig.3c/S9**, the 'loadings' would correspond to the values each gene contributes to the PCs. However, **Fig. 4c/S8** suggests response and input variables have been swapped. Please clarify. The main text further states that 'Strikingly, PCA analyses of genes identified by differential intronic read signal after each epigenetic perturbation resulted in similar principal components as the analyses of new RNA'. It is not clear what precisely is meant by 'similar principal components' or how **fig. s8** demonstrates this. furthermore, the compound labels in fig. 4c are too small.

We are deeply grateful to the reviewer for the attentiveness in identifying an error in the initial version of the manuscript. The revised manuscript has addressed these comments with several changes. Firstly, the values in the PCA heatmaps represent PCA scores, not loadings as previously and mistakenly stated. Secondly, the heatmaps are organized by compound class, rather than by distances derived from hierarchical clustering. This clarification, along with a description of how the principal component decomposition was performed, has been added to the methods section under the heading "Principal component analyses." The figure legends for Figures 4 and S8 have been revised to reflect these changes, and figure labels have been enlarged to aid interpretation.

The similarities in response between the observed signals from intronic reads and 4sU-based new RNA indicate that the same compounds are separated with similar directions and relative effect sizes in the respective PCA plots of the two datasets. The primary responding compounds following treatment are the hydroxamic acid HDACs and a subset of bromodomain inhibitors (NVS-1, PFI-1, SGC-CBP30, and bromosporine). This signal is detected in both new RNA and intronic reads, as demonstrated by the similarities in Figures 4 and S8. The main text has been updated on page 9 to clarify this point.

Fig. 1e is interpreted as a demonstration that total RNA masks the SAHA effects. An alternative interpretation would be that the new RNA does not capture single cell variation, although I grant that that is unlikely and the observation rather reflective of sensitivity issues. But perhaps it is still worth discussing? Also, is it worth discussing the 'new RNA' plot in fig. 1e as a justification for doing bulk rna-seq, given that the treatments lead to fairly homogenous subpopulations?

We addressed this point by measuring cell-to-cell expression correlation in new and total RNA. There is a significantly higher correlation in total RNA compared to new RNA (Reviewer Figure 3, pasted below, and new revised **Figure S1d**). This is expected since the new RNA signal is a smaller temporal snapshot of recent transcription with fewer overall RNA molecules counted, and it is therefore more noisy for both biological (transcriptional bursting) and technical reasons (sampling across fewer RNAs in library preparation). Despite the increased noise, we show throughout the manuscript that the new

RNA profiles capture treatment-responsive genes that would otherwise be masked in the total RNA signal. The reviewer correctly notes that this further supports the validity of bulk-RNA screenings.

Reviewer Figure 3 (revised manuscript Figure S1d). Cell-to-cell variation in new and total RNA profiles.

p6, first paragraph: how were the binomial and chi square tests carried out? What exactly was tested, what were H0 etc? same for Anderson-darling on **p8** (why wasn't a Mann-Whitney used?)

Thanks for this comment. For the binomial and chi square tests, we have added this information to the methods section "Inference of transcriptional bursting from new RNA profiling of individual cells". The binomial test (for Figure 2c) evaluated the obtained numbers (12 and 44) in the figure panel, against the null hypothesis being equal chance for both modes of expression change. The chi-square test evaluated whether (observations: 12, 44, 9 and 19 in the same panel) against the null hypothesis of equal ratio between the two modes for the two regulatory directions.

Yes, we changed the Anderson-darling test for the the commonly used Kolmogorov-Smirnov test for cumulative distributions and obtained highly similar p-values. We updated the result section accordingly.

p19: it is stated that the prior for π_g is 0.1 – shouldn't this be a probability distribution rather than a fixed value if MCMC sampling is used?

Thank you. The prior that is used is a uniform distribution between 0 and 1, with an initial guess of 0.1. This error was corrected in the revised version of the manuscript.

Fig. S1: Shouldn't the number of newly expressed genes be close to zero without 4sU if the inference is working correctly?

The reviewer is correct, and we also found this figure puzzling at first. We believe it partly stems from genetic variation in K562 cells that were not fully removed (i.e. SNPs affecting the same base conversion as 4sU), and from noisy inference due to genes with low read numbers. As shown in Supplementary Figure 3, simulations show that read numbers have an effect on the accuracy of the inferred fraction new RNA. At low read numbers, the inference becomes unstable and true zeros are confidently called as having new reads. Possibly, we could have limited analyses to more highly expressed genes, yet we did not see this background impacting any of the conclusions of the manuscript. The background in new RNA signal would negatively impact our ability to find true changes in new RNA profiles, yet that was definitely not the case as shown throughout the manuscript. Finally, the observation that intronic read-based analyses identified very similar compound-specific direct transcriptional effects further demonstrated that the background did not limit the conclusions. In the revised manuscript, we clarified this background in the figure legend for Fig S1 through this statement: "(b-c) The residual background of new RNA observations for samples not subject to 4sU

likely comes from genetic polymorphisms and instable inference for low-expressed genes (as shown in Supplementary Figure 3).”

Reviewer #2 (Remarks to the Author):

The manuscript by Hartmanis et al. describes an updated workflow (termed NASC-seq2) for SLAM-seq based nascent mRNA profiling for characterizing cellular responses to drug perturbations. The major advance lies in the implementation of Smart-seq3-like nanoliter liquid handling technology in a 384-well format, which substantially increases the scalability of the assay, e.g. for compound screening applications. The authors first use NASC-seq2 to measure direct effects of SAHA/vorinostat-mediated HDAC inhibition on transcriptome-wide mRNA synthesis in single K562 cells on a scale of hundreds of cells per condition. The authors next apply NASC-seq2 to “mini-bulk” K562 populations (10k-100k cells/condition) treated with an expanded panel of 28 compounds annotated as HDAC inhibitors. This experiment resolved some structure-activity-relationships within the compound panel. Finally, the authors tested 55 more epigenetic-focused compounds (+ the 28 HDACi) and could again stratify compound classes based on acute transcriptional phenotypes with moderate success.

*Overall, the manuscript is clearly structured and well written, and the provided methods (including a detailed protocol on protocols.io) are of sufficient detail to implement NASC-seq2. The presented analyses are technically sound and sufficiently support the presented conclusions. The finding that intron-based analyses from total RNA provides similar information content as SLAM-seq based profiling – at least on a ‘mini-bulk’ level – is interesting and broadly relevant. Nevertheless, NASC-seq2 could serve as a useful tool for medium-throughput screening endeavors with transcriptional readout at high kinetic resolution. However, besides describing a technical advance that increases the scalability of nascent mRNA profiling, the manuscript **lacks conceptual and technical novelty and provides very little mechanistic insight**. Many of the presented conclusions (e.g. concerning differences between total and nascent mRNA profiling) are well-described in the published literature. Moreover, SLAM-/NASC-seq profiling has been previously applied in low-input settings by the authors and others, and the presented profiling of 83 compounds in K562 cells does not fully demonstrate the utility of NASC-seq2 at scales that are relevant for compound screening.*

We thank the reviewer for all comments on our manuscript. As shown in the revised manuscript, we have reworded the text throughout to reflect that we have adopted existing procedures into a more automatable and scalable implementation, which allowed us to profile larger numbers of compounds. We don't agree with the criticism against conceptual and technical novelty and little mechanistic insights, we do find the technical novelty obvious since we removed many costly and time-consuming steps that enables us to in parallel perform hundreds of SLAM-seq equivalent experiments per day. Additionally, our study provides conclusive evidence demonstrating that intronic reads in standard RNA-seq capture the same information as using 4sU (30 or 60 min). We believe this is a significant statement that will greatly simplify experimental procedures for capturing direct transcriptional effects.

The following are my major concerns:

Both conceptually and technically, NASC-seq2 is largely based on ideas, innovative chemistry, and protocols that were originally described as SLAM-seq, which is only referred to once as “SLAM-seq chemistry”. At several points (including the abstract), the authors do not properly distinguish between prior art and technical advances provided by their manuscript. Similar is true for conceptual findings and conclusions. For example, **Fig. 1** mainly describes differences between nascent and total mRNA-seq that (i) have been well-described using SLAM-seq and other nascent RNA-seq tools, and (ii) are unsurprising given the short sampling time points in case of total RNA-seq. **In general, the authors should give proper credit to previously established concepts and methods and more clearly describe the specific improvements of these methods offered by their manuscript.**

We apologize that the initial manuscript came across as not properly citing crucial previous literature. That was not our intention. We note that the initial manuscript contained strong sentences in the introduction giving credits to SLAM-seq and its use for genetic perturbations, e.g. in these sentences:

“Pioneering studies have demonstrated that direct transcriptional effects of pharmacological or genetic perturbations are visible in nascent RNA, minutes to hours after perturbations²⁻⁴.” Referencing both original SLAM-seq paper (ref 3) and the pioneering identification of the BRD4-MYC axis (ref 4).

“Specifically, our approach builds on the SLAM-seq chemistry³ that allows for in silico separation of labeled (new) and unlabeled (old) RNA through the induction of sequence errors during reverse transcription of labeled RNA molecules.”

We have taken the reviewers comment seriously and therefore reworded the text throughout the manuscript to more clearly pinpoint the areas of innovation both in terms of methodology and insights, the largest impact of the mini-bulk 4sU implementation is that we can dilute out the alkylation reagent (instead of clean-ups), removed several unnecessary washing steps that enables analyses of fewer input cells. Moreover, the implementation is automated in 96-well plates in pipetting robots. We do believe the scale achieved in this manuscript is 10 to 100-fold higher than any SLAM-seq based study to date. On the computational side, we provide an open-source implementation to the new RNA inference that has been optimized for NVIDIA GPUs.

In the revised manuscript, we have reworded the text throughout to better tune the improvements and to double check that proper credits have been given.

While NASC-seq2 could provide an important step towards the use of nascent RNAseq for compound screening, the manuscript **does not fully establish the scalability and broad applicability of this approach**. All presented ‘mini-bulk’ assays were performed using suspension cells (K562), which can be cultured at much higher densities in multi-well formats compared to adherent cells, which are the more relevant system for compound screens. The **authors should demonstrate the feasibility of their 384-well assay in adherent cells**. In addition, 83 compounds seems a rather limited number for screening applications overall, and at this scale, more interesting compounds could have been selected to demonstrate relevant ‘use-case-scenarios’ of their assay. For example, NASC-seq2 could be used to ‘de-orphanize’ the mechanism of action of novel compounds, to identify and characterize off-target activities, or to benchmark on-target potency. None of this is convincingly shown.

The reviewer asks for more general experiments that can demonstrate larger use cases of the new RNA based paradigm for mapping the direct effects of therapeutic compounds. Our experiments were performed on suspension cells (K562) and expanding the study to adherent cells would showcase stronger utility on cells more routinely used for compound screening.

We note that performing a new experiment, involving a new cell type and acquiring a different panel of cell-line relevant compounds is a time-consuming endeavour, yet we followed the reviewers suggestion and constructed a large new experiment on the adherent breast cancer cell line MCF7. For this experiment we selected a completely new set of 58 breast cancer relevant compounds (complete list added as **Supplemental Table 2** in revised manuscript) that was subjected to MCF7 cells at different doses. For each compound, we treated MCF7 cells at five different concentrations spanning from 1 μ M to 10mMs for 4 hours, and constructed mini-bulk libraries that were sequenced. Together with nine untreated control MCF7 cell preparations, this allowed us to fit dose response lines for each gene and drug, with an associated p-value for responding to the drug (example in **new Supplementary Figure 10a, pasted below**). At least one significant gene was found for 13 of the drugs after multiple testing correction, at 5% false discovery rate (**new Supplementary Figure 10b, new Supplementary**

Table 3). To identify affected pathways and functions, we conducted a gene ontology overrepresentation analysis. This method, while needing more than just a few genes, is robust against random false positives. Consequently, we adopted a higher false discovery rate (50%). This approach corroborated²² that phosphoinositide 3-kinase and mammalian target of rapamycin inhibitors impact translation (**Reviewers Figure 5, pasted below, new revised manuscript Figure 4j**). No other drugs or gene ontology terms remained significant after correcting for multiple testing for gene ontology terms. To prevent bias favoring highly expressed genes, we employed two strategies: comparing down-regulated genes against up-regulated ones, which should balance out the bias, and using expression-matched background gene sets. Although doxorubicin, a DNA-damaging drug, significantly reduced the expression of numerous genes (as shown in **Supplementary Figure 10b**), there was no significant pattern observed in any gene ontology terms, indicating a non-specific response to DNA damage at our time point of analysis.

j

Drug	(Drug target)	2-stageBH FDR within comparison			Comparison
Alpelisib	(PI3K)	0.00013	0.00013	0.00085	Down, expr. matched
Apitolisib	(PI3K/mTOR)	6.1e-08	2e-07	8.1e-08	Down, expr. matched
Apitolisib	(PI3K/mTOR)	0.0012	0.0012	0.0012	Down against up
Buparlisib	(PI3K)	0.0028	0.0028	0.0077	Down, expr. matched
Temsirolimus	(mTOR)	0.026	0.046	0.026	Down, expr. matched

10^0 10^{-2} 10^{-4} 10^{-6}
 2-stageBH FDR

structural constituent of ribosome translation ribosome
 GO term

Reviewers Figure 5 (revised manuscript new Figure 4j). Gene ontology enrichment of compound-responsive genes identified from new RNA profiling of MCF7 cells. We used two different methods to avoid sensitivity bias due to expression level: matching genes by expression rank or comparing downregulated to upregulated genes. *P*-values are corrected for multiple testing within each of the resulting in 128 comparisons for 40 compounds (the additional 18 compounds did not pass quality-control filtering due to insufficient samples per compound), with the significant results being shown.

Reviewers Figure 6 (revised manuscript new Supplementary Figure 10). Compound screen in MCF7 cells. (a) An example of the statistical test employed: least squares linear regression of five concentrations plus nine negative controls that are the same for the comparison with every drug, with a Wald test of a t-distribution (scipy.stats.linregress). Due to the log-uniformly distributed concentrations and the presence of zeros, we have $\log_{10}(1+\text{conc})$ transformed the x-axis and $\ln(1+\text{count})$ transformed the y axis prior to regression, as the highest concentration alone would otherwise drive the result. The line shows the fit from the same python function as the p-values come from. The gene is considered downregulated in this treatment since the slope is negative. **(b)** A list of compounds and the number of significant genes at a 5% and 50% FDR cutoff. The 50% cutoff is later used for gene ontology since that analysis method handles noisy input.

By providing this new experiment we show that new RNA profiling – using the mini-bulk strategy developed in the manuscript – apply to adherent cells and to compounds that act on cellular signaling processes we provide scalability and broad applicability.

The manuscript provides little mechanistic insight into chromatin regulators or the investigated compounds. The presented ENCODE ChIP-seq analyses remain correlative at best and do not yield conclusive insights. Compound selection is questionable at times, with multiple non-selective, broad-spectrum inhibitors included (e.g., bromosporine, nicotinamide, phenylbutyrate), while more selective chemical probes could have been used to study sub-class specific effects (e.g., Merck60, Entinostat; see e.g., PMIDs 28228643; 35484434), or clinical-stage compounds could have been included to add relevance (e.g., panobinostat, entinostat, pelabresib/CPI-0610, apabetalone/RVX-208). The moderately successful stratification of compound classes in **Figures 3 and 4** is likely due to the below-par compound selection, rather than NASC-seq2 technology as such. Fewer but better chemical probes per epigenetic class – perhaps in combination with degron-engineered cell lines – would likely be required for deeper mechanistic insights. Conversely, applying NASC-seq2 to different cell lines or even (patient?) PBMCs exposed to a panel of clinical-grade compounds would add real-world relevance and leverage the single-cell aspect of the technology.

We acknowledge the reviewer's point regarding the extensive range of compounds available for testing their direct transcriptional effects. However, we believe that this manuscript represents a significant advancement, as it assesses the impact of the first 83 epigenetic compounds on transcription after 30 and 60 minutes. This study highlights the distinct insights gained when analyses are conducted in both single-cell and mini-bulk formats.

In response to the suggestion of expanding our methodology to other cell types, we have indeed extended our scope to also include adherent MCF7 cells. This involved screening an additional 58 compounds relevant to breast cancer treatment, which revealed notable compound-specific transcriptional effects in the new RNA profiles. While introducing more compound analyses could uncover further transcriptional variations, expanding our study to include another set of epigenetic compounds is beyond the current manuscript's scope.

Minor concerns:

The 'up-regulated' axis in **Figure 4a (right half)** seems stretched (also the outlier dots seem stretched in the SIRT-activator and HAT-inhibitor boxplots). Please double-check that there was no mishap during figure preparation.

Thanks, fixed.

The axis labels, especially for the ENCODE ChIP-seq volcano plots and all log₂ fold-changes, do not clearly convey what is being compared to what. Please consistently indicate which groups were compared (e.g. 'ChIP-seq peak enrichment log₂[C4 vs. C1-3 transcripts]', ...).

Fair point. We have changed the figure labels and legends to clearly state which cluster(s) are being compared in each such figure item.

The value of clustering into exactly 3-4 clusters (**Figures 3 and 4**) is unclear. Why perform ENCODE ChIP-seq analysis exactly for C2 and C4 in **Figure 3d**? If analyses are primarily focused on strongest up- vs. down-regulated genes, why not cluster into fewer clusters (e.g., merging C1/C2) or add a log₂ fold-change cutoff averaged on the strongest compounds (i.e. the hydroxamic acid derivatives)

We opted for a hierarchical clustering based partitioning of genes, since it is transparent and data-driven. However, as the reviewer points out, the cutoff for defining clusters has to be set and affects the numbers of clusters obtained. We had conducted ChIP-seq analyses using many variations of cluster sizes at different cutoffs, and they yielding very similar results, and we simply chose reasonable cutoffs, while acknowledging that one or more clusters could be merged.

We however decided to update Figure 3 by merging old clusters C1 and C2 into one cluster, since we agree with the reviewer that they indeed are very similar and that representation is perhaps more concise. The results, presented in new Figure 3d-f, are very similar to the previous ones.

Were spike-ins used to control for potential global changes in RNA synthesis and if so could you please comment on why you did not deem it necessary in this case?

While spike-ins can serve as a valuable control for variations in global RNA synthesis rates, we concluded that they were unnecessary for our study. In the single-cell experiment, we counted RNA molecules per cell and found no differences in total RNA molecules present per cell, and that the new RNA counts increased with labeling time (**Figure 1d**), as expected. Therefore, we similarly did not use them in the mini-bulk experiments since there is no reason to believe that the total RNA amounts would differ (i.e. the pre-existing RNAs should be constant irrespective of any treatment). Moreover, the amount of new RNA per gene is calculated as a fraction of the gene's total expression level and as we show in Figure 4a we do see that the fraction of new RNA can be significantly impacted by certain compounds, in particular bromodomain inhibitors.

There seems to be a typo in the main text linked to **Fig. 3e**. The text says “Conversely, binding of the same factors was found depleted in promoters of the genes that were downregulated after SAHA treatment (**Figure 3e**).” We believe **panel 3e** focuses on Cluster C2, which is the genes mostly upregulated after SAHA treatment?

Thanks, the typo has been fixed in the revised manuscript.

The panel of compounds labeled as “BET inhibitors” would more accurately be described as a panel of “Bromodomain inhibitors”, as e.g. SGC-CBP30 is an inhibitor of the CBP/p300 bromodomain with ~100-fold selectivity over BET protein (BRD2/3/4/T) bromodomains. This should also be changed in the main text (**page 8-9**)

Fixed.

Reviewer #3:

Summary: In this manuscript, the authors developed an updated version of NASC-seq (NASC-seq2) to profile newly transcribed RNA from single cells or mini-bulk samples. They applied this method to measure the direct transcriptional effects of inhibitors targeting epigenetic regulators. The present manuscript is potentially important because the proposed method can be a building block of a platform for large-scale chemical screening quantifying direct transcriptional effects. However, there are several important points that should be addressed to substantiate the authors' claims.

Major points:

In **Figure 1e**, the single-cell transcriptomes appear to be largely affected by the 4sU labeling times, potentially contributing to batch effects. The authors should provide at least two replicates for each condition, and examine this issue carefully.

The transcriptomic differences between cells at various 4sU labeling times are expected. In 4sU experiments, cells consistently separate based on the duration of 4sU exposure, as different exposure periods typically lead to different gene sets becoming expressed. At the 30-minute mark, double the amount of RNA is identified, as new RNA transcription occurs over time, as we showed in this experiment in **Figure 1d**.

We note that this experiment is internally controlled, since i) no difference in overall new RNA signal with or without SAHA was observed (**Figure 1d**) and ii) no separation between any conditions is detectable at the total RNA level. Yet, large transcriptomic differences were captured in new RNA profiles specifically, as visualized in the tSNE (**Figure 1e**). Finally, the gene sets identified in the single-cell experiment was consistently identified in the mini-bulk experiments that were performed with many biological replicates per compound (**Figure S4a-c**).

In Supplementary **Figure 1d** and **e**, as discussed by the authors, the intronic reads seem to be underrepresented in single-cell NASC-seq2 data and have a limited statistical power to identify differentially expressed genes in terms of new RNA. However, this was not observed in mini-bulk NASC-seq2 data (Supplementary **Figure 4g**). The authors should carefully examine this issue by investigating the potential causes of this issue (e.g. 3' bias of read coverage within a transcript, low sensitivity).

We investigated the reasons behind the lower intronic signal in our NASC-seq2 single-cell data compared to our mini-bulk samples. We've identified two key factors, which are detailed in **Supplementary Figure 4h** of the revised manuscript (also pasted below). First, the addition of 4sU labeling slightly reduces the number of intron sequences we recover. Second, there is a stronger protocol-specific effect, as the ratio of intronic to exonic mapping reads is eight times higher in our mini-bulk data. Moreover, we performed a RNA bias analysis to compute TIN values (sometimes used to infer RIN values, reference: Wang et al. 2016, reference 30) which demonstrated lower TIN/RIN values for the single-cell 4sU-based method. Interestingly, other full-coverage single-cell RNA-seq methods, e.g. Smart-seq3xpress (reference: Hagemann-Jensen et al. 2022) show intron coverage similar to the mini-bulk. We believe that the bias in the single-cell data comes from the fact that the 4sU incorporation and alkylation reactions have negative impact on the RNA which lowers the efficiency with which reverse transcription can generate full-length cDNAs, leading to read coverage with fewer introns and a higher 3' end signal.

Supplementary Figure 4h. Quantification of intron-coverage and RNA integrity in single-cell and mini-bulk experiments. Boxplots showing per-sample ratios of reads mapping to intronic versus exonic regions. Notches show bootstrapped 95% confidence intervals. We calculated transcript integrity (TIN) numbers, a computational measure of 3' bias that translate into approximate RIN numbers³⁰, which revealed lower TIN/RIN numbers for the single-cell data.

In **Figure 3**, it would be more informative if the authors provide a list of genes showing primary or secondary downstream effects, and discuss the biological differences of the two sets of genes.

We appreciate this comment and while we believe that a distinction between primary and secondary transcriptional effects is highly relevant – and at the core of the message of this manuscript – a precise definition and capturing of both is harder to achieve. One could argue that a system with two distinct labels supplied to cells at different time points could help answering this question, a direction we tried but unfortunately failed to make progress in. In such an experiment, new RNA can be divided into groups based on the labeling time, informing on what genes are actively transcribed per time period. Moreover, depending on the compound administered to cells, transcriptional changes may occur already within a few minutes after stimulation (e.g. as shown in TT-seq, Schwalb et al. 2016) making it hard to discern when primary effects end and further downstream effects begin.

In this study, we used 30- and 60-minute drug treatments and the responsive-genes to share specific features (binding of specific factors to their promoters, histone marks) that makes us confident that this set is (at least) significantly enriched for primary effects. Detecting transcriptional differences at 6 or 24 hours post-treatment, the standard procedure of many systematic efforts to date, essentially only capture secondary effects as we show in Supplementary Figure 6.

In the revised manuscript, we have provided supplementary tables that list significant compound-responsive genes detected at the new RNA level for each compound.

In **Figure 4**, the authors hypothesized that the enrichment of YY1 binding in the promoters of downregulated genes can be explained by the rapid acetylation of YY1 after SAHA treatment. This prediction should be experimentally validated.

The reviewer highlights our original discussion on YY1 status after SAHA treatment, which was poorly worded: “It is plausible that SAHA treatment rapidly acetylates YY1”. Our intended meaning was that HDAC inhibitors, like SAHA, through its inhibition of HDACs will de-repress their targets, since YY1 has been identified as a target of HDAC1 (e.g. Sheng Dong et al 2017, reference 25 of the revised manuscript). In the revised manuscript we have improved the phrasing of this discussion to clarify the intended meaning.

The authors should discuss the key features of NASC-seq2 by comparing it with NASC-seq.

We appreciate the comment. The development of NASC-seq2 was the focus of another study from our lab (Ramskold et al. Nature Cell Biology 2024, reference 5) and in that study we detailed the experimental modifications and benchmarked the improvements with NASC-seq2 over NASC-seq. In that study, transcriptional bursting was more generally examined through the temporal precision obtained using through the 4sU RNA labeling. Therefore, the development of NASC-seq2 was not included in this manuscript. In this study, we used NASC-seq2 to study direct transcriptional effects of perturbations that can be captured by monitoring new RNA profiles. In the revised manuscript we have clarified these relationships throughout the manuscript.

Minor points

“signal-to noise ratios” should be “signal-to-noise ratios” at page 6.

Thanks, fixed.

Supplementary Figure 6f referred to at page 8 does not exist.

Fixed (should refer to Supplementary Figure 6e) in the revised manuscript.

“H₀: p=p_c” at page 14 should be “H₀: p=p_e”.

The reviewer is correct, the typo has now been fixed.

At page 18, “algorithm described in Jürges et al11” should be “algorithm described in Jürges et al18”.

The reference has been updated to refer to the correct publication.

REVIEWER COMMENTS

Reviewer #1 (Remarks to the Author):

The authors have made a good effort to improve the manuscript in response to my comments, although some of it seems to have been done in a hasty way and would benefit from further efforts, as explained below.

- Text explaining the bursting inference has been added but needs proofreading and correction (e.g. '[...] need its inverse since know the count distribution [...], but kon or ksyn'). A ref to the authors' previous paper (Ramskoeld et al, Nat Cell Biol '24) on this has also been added, but it should be noted that bursting inference on 4sU single cell data was to my knowledge first described by Edwards DM et al, Genome Biol, '23, so I'd appreciate if this ref could also be added (also since it is not in the Ramskoeld paper either).

We apologize for making this section confusion and incomplete. We have clarified the method section with only the critical information that specify this particular experiment, and otherwise refer to Ref 5 as we follow that methodology and code.

- according to the rebuttal, the bursting inference section is also supposed to contain information on the statistical tests, but it does not.

We have added information on the permutation tests performed, and that the resulting P-values undergo a Benjamini-Hochberg correction, using a 5% false discovery threshold.

In particular, it is still not clear to me why a KS test (was Anderson Darling in the original manuscript) was used instead of e.g. a Mann-Whitney, when the authors seem interested in showing higher HDAC binding, i.e., a shift in mean or median – a KS/AD could show significance due to different shapes of the distributions even when the median is equal, I think. This is a minor issue, though; I believe the authors' claim of higher HDAC binding.

Replaced with Mann-Whitney test, which only strengthened the conclusions with lower P-values, the revised text now reads:

"HDAC1 and HDAC2 binding to induced genes was significantly more pronounced in new RNA profiles after 1- and 3-hour SAHA treatments (with 1 hour 4sU exposure) compared to total RNA profiling at 6 and 24 hours ($P < 0.001$, Mann-Whitney U test) (**Figure 3g**)."

Further minor issues:

- the compound labels in fig. 4c are still tiny, but perhaps this is ok if readers can zoom in on a pdf

We found no better solution, still wanted to show the full names in the figure.

- my PCA queries have been resolved, but the claimed similarities among the new RNA PCA and intron PCA are hard to see, given that fig. 3c only shows HDAC inhibitors and fig S9 all (and with similar blue/green/cyan shades for different groups), and the heatmaps (fig4c, fig.s8) don't appear similar either, apart from the hdac inhibitor 'block' high in PC1 and PC2 in fig. s8 and fig. 4c, respectively.

Fair point. We agree with the reviewer that the similarity was not systematically explored, rather we reached that conclusion after spending time digesting these figures.

We opted for making the similarities between the two PCA heatmaps more systematic, and therefore we correlated each PC component against each other, i.e. each PC component derived for new RNA profiles against each PC component derived from the intron analysis. Those correlations are shown in the heatmap to the right below (also added as revised Figure S8b). In addition, we used linear regression to evaluate whether each PC component could be predicted as linear combinations of PC components of the other type, i.e. for PC1 of Intron reads (left column), this component could be well predicted (98%) based on the PC of new RNA profiling. To the right below we show a control detailing what correlations and linear regression results can be due to chance.

Revised Figure S8b,c. (left) Correlations between the principal components for intronic reads and those in Figure 4c for new RNA. R^2 values (explained variance) from linear regression of one principal component against the 10 principal PCs in the other set are shown in parentheses at the edges. These values illustrate how many intron-data principal components are simply linear combinations of nascent RNA principal components, rather than truly differing. (right) Like (left), but with genes shuffled, demonstrating the sizes of random correlations between principal components, reaching no higher than $r = 0.27$ for a fit to one principal component and $R^2 = 19\%$ for a fit to 10 principal components, in contrast to the larger numbers in (left).

Our conclusion is that, although the order of principal components vary, sprinkled with assay specific noise, the main grouping of compounds, e.g. hydroxamic acid HDACs and a subset of the bromodomain inhibitors represent the main PC components identified in both analyses. With this added analysis we believe the similarity between experiments are more directly shown.

Reviewer #2 (Remarks to the Author):

In their revised manuscript, Hartmanis et al. have comprehensively addressed all my questions and concerns through text editing and extensive additional experiments. I appreciate that conducting an additional screen of 58 compounds in MCF7 cells has been a major endeavor, and I would like to thank the authors for this effort. Together with the original screen in K562 cells, this screen in an adherent cell line, which includes a well-selected panel of compounds, provides a valuable pilot and benchmark for future applications of this technology. I have a few remaining comments and suggestions regarding the presentation of these data:

1. Several of the investigated compounds (e.g. MEKi and others) must be expected to trigger clear transcriptional responses in MCF7 cells after 4 hrs. In this light, the reported effects are surprisingly limited. The authors state that they identified at least one significant gene for 13 of the drugs-For the two MEKi they only identify 2 response genes for Trametinib and none for Cobimetinib, which is far below the expected effect size. The authors should comment on this. I fully agree that this problem might reflect overly stringent statistical tests, and the presented relaxed analysis indeed yields effects sizes in the expected range.

Fair point. We think it may be due to two experimental parameters we chose, e.g. that we performed dose-response curves that include fewer higher concentration perturbations, and that our dose-response curve had in general lower concentrations of the compound than the replicates used for epigenetic compounds. These are only speculations at this point. We added a new analysis in response to the question below, that further supports that only a sub-set of the compounds used had a strong observed perturbation in the new RNA profiles at the chose time period.

2. To gain confidence in the assay and data, the authors should analyze and present whether functionally related compounds (i.e. inhibitors that hit the same target) lead to similar responses. The presented GO term analysis does not really address this point, and it would be most informative to perform hierarchical clustering or PCA analysis over the entire dataset (under relaxed statistical stringency).

We complemented the analysis of the manuscript, with hierarchical clustering over the entire dataset to better explore and visualize the overall patterns of compound perturbations. In the figure below, also added as Revised Figure 4j, it is visible that certain compounds group according to their class of targets, e.g. the PI3K, AKT and mTOR inhibitors, EGFR and Topoisomerase inhibitors, whereas certain compounds fail to show similarities with other perturbations.

New Figure S4j. Average linkage hierarchical clustering of a breast cancer MCF7 compound screen based on fitted dose-response slopes for 3,479 genes, where all drugs yielded a slope fit. Colors indicate an $r = 0.33$ cutoff, highlighting the PI3K/AKT/mTOR inhibitor class. Drug targets are shown in parentheses.

This added analysis complements the drug specific Gene Ontology results.

3. While the presented slope analysis over different doses is intriguing, this is only one of many ways to analyze the data. Alternatively, the authors could perform analyses (e.g. drug comparisons described above) with just one of the higher doses. Importantly, to enable readers and potential users to perform such analyses, the authors should provide all primary data of both screens (i.e. primary NGS read counts) as Supplemental Information.

Definitely. We have deposited all raw data (FastQ files) and also count matrices (new and old RNAs) in ArrayExpress to facilitate re-analysis efforts to test additional strategies to gain further insights from this data.

Once more, I would like thank the authors for an excellent revision.

Thanks for good comments throughout the revision.

Reviewer #3 (Remarks to the Author):

The authors have satisfactorily addressed my comments.